# 3D Gaussian Flats:
# Hybrid 2D/3D Photometric Scene Reconstruction

**Maria Taktasheva**
Simon Fraser University
maria_taktasheva@sfu.ca

**Lily Goli**[*]
University of Toronto

**Alessandro Fiorini**
University of Bologna

**Zhen Li**
Simon Fraser University

**Daniel Rebain**
University of British Columbia

**Andrea Tagliasacchi**[*]
Simon Fraser University
University of Toronto

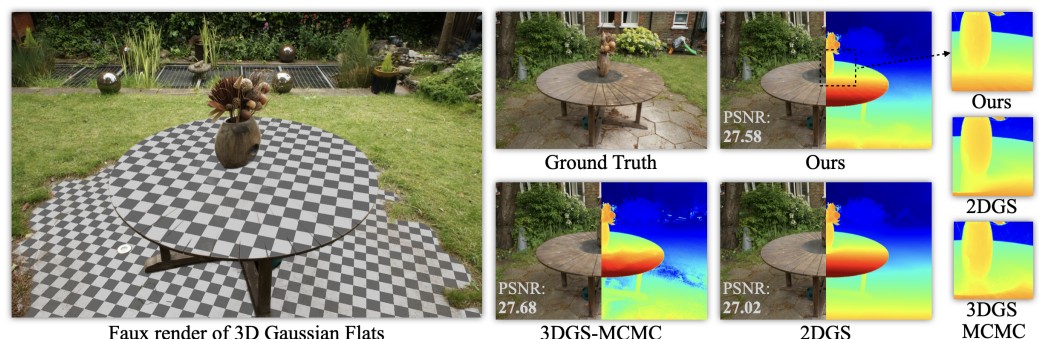

Figure 1: **Teaser** – We introduce 3D Gaussian Flats, a hybrid representation of 2D Gaussians on semantically distinct planar surfaces and 3D Gaussians elsewhere (left). Our method achieves a photorealistic quality on par with fully 3D approaches, while improving geometry over surface reconstruction methods (right) e.g. no visible hole in the middle of the 'garden' scene from Mip-NeRF360 [1].

## Abstract

Recent advances in radiance fields and novel view synthesis enable creation of realistic digital twins from photographs. However, current methods struggle with flat, texture-less surfaces, creating uneven and semi-transparent reconstructions, due to an ill-conditioned photometric reconstruction objective. Surface reconstruction methods solve this issue but sacrifice visual quality. We propose a novel hybrid 2D/3D representation that jointly optimizes constrained planar (2D) Gaussians for modeling flat surfaces and freeform (3D) Gaussians for the rest of the scene. Our end-to-end approach dynamically detects and refines planar regions, improving both visual fidelity and geometric accuracy. It achieves state-of-the-art depth estimation on ScanNet++ and ScanNetv2, and excels at mesh extraction without overfitting to a specific camera model, showing its effectiveness in producing high-quality reconstruction of indoor scenes.

[*]Equal Advising

39th Conference on Neural Information Processing Systems (NeurIPS 2025).

# 1 Introduction

Recent advances in radiance fields and novel view synthesis have enabled the creation of realistic digital twins from collections of real-world photographs [2, 3]. These techniques allow for high-fidelity 3D reconstructions that capture intricate details of real-world scenes, making them invaluable for applications in virtual reality, gaming, cultural heritage preservation, and scientific visualization.

However, when optimizing for novel view synthesis on flat and texture-less surfaces (e.g. walls, ceilings, tables that are prevalent in indoor scenes), current methods struggle in producing a faithful 3D reconstruction as the problem is photometrically under-constrained [4]. Specifically, modern novel view synthesis frameworks like [5, 6], which are optimized via volume rendering, model flat surfaces with low densities, resulting in non-opaque representations of solid surfaces; see the surface of the table in Figure 1 as an example. Conversely, surface reconstruction methods that assume solid, flat surfaces avoid this limitation [7]. However, they compromise visual quality in favor of a more parsimonious 3D reconstruction; see figure 1. Our core research question is whether these seemingly conflicting objectives could be achieved simultaneously.

Some approaches have attempted to answer this questions by first creating a full 3D representation, and then – *post-training* – detecting planar surfaces to enable 3D planar reconstruction [8, 9]. However, these methods do not leverage planar assumptions during the optimization of the scene representation itself, limiting their effectiveness. Others enforce planar assumptions during training through various regularizer losses [10]. However, these losses can be hard to tune, as they are only suitable for the portion of the scene that is solid and flat, hindering the reconstruction whenever these assumptions are violated.

In contrast to these methods, we propose to look at the problem in an *end-to-end* fashion, conjoining the process of photometric to the one of planar surface reconstruction. To achieve this, we introduce a *hybrid* 2D/3D representation, where flat surfaces are modeled with 2D Gaussian splats [7] that are confined to 2D planes, while the remaining of the scene is modeled with a classical, and more expressive, 3DGS model [6]. By *jointly* optimizing planar (2D) and freeform (3D) Gaussians, our approach enables better fitting of the final representation to planar surfaces within the scene. During photometric optimization, our method dynamically detects planar regions, and adaptively grows their extent, resulting in reconstruction that *retains* high visual quality (as measured by PSNR) compared to a classical 3DGS scene, while simultaneously achieving superior geometric accuracy (as measured by depth error).

Our evaluations demonstrate that this hybrid representation achieves state-of-the-art depth estimation results on challenging indoor datasets including the new ScanNet++ dataset which was designed for dense reconstruction tasks using NeRF-based approaches, and the legacy ScanNetv2 dataset with sparser camera views. Our method delivers crisp reconstructed surfaces, while maintaining competitive visual quality compared to fully 3D representations. Beyond novel view synthesis, our approach has application in mesh extraction for planar surfaces, producing high-quality meshes and accurate mesh segmentation results across diverse capture setups (DSLR and iPhone captures), without the overfitting issues that negatively affect previous methods trained on specific camera models.

# 2 Related Work

Modern neural scene reconstruction methods aim to generate high-quality 3D representations from 2D images for applications like novel view synthesis [5, 6]. Despite significant progress, volumetric approaches struggle to accurately reconstruct planar surfaces [11], while surface reconstruction methods fail to recover volumetric effects [12]. Finding an approach that accurately reconstructs planar geometry without compromising the quality of the surrounding scene geometry and appearance is a key challenge.

**Representations for differentiable rendering**    Neural Radiance Field (NeRF) [5] pioneered scene reconstruction with a 3D neural representation optimized through differentiable volumetric rendering. 3D Gaussian Splatting (3DGS) [6] overcame NeRF slow training/rendering speed by representing scenes as efficiently rasterizable 3D Gaussians, dramatically accelerating rendering while maintaining quality. The impressive speed-quality balance of 3DGS quickly established it as a standard approach,

with recent advancements such as 3DGS-MCMC [13] further enhancing its accessibility by eliminating the dependency on SfM initialization. Despite these innovations, volumetric representations still struggle with clean geometry reconstruction in flat and textureless surfaces common in indoor environments, hindering applications like mesh extraction. Our method addresses these challenges through a hybrid 2D/3D Gaussian representation that achieves superior geometric reconstruction while preserving rendering quality.

**Surface representations and planar constraints**   While NeRF [5] and 3DGS [6] employ fully volumetric representations, alternative approaches such as [11, 14] model scenes as solid surfaces. This philosophy inspired SuGaR [15], to use a regularization term that encourages the Gaussians to align with the surface of the scene, and later 2DGS [7], which uses 2D Gaussian primitives to reconstruct surfaces outperforming prior surface reconstruction methods [11, 14, 15]. Recent work [16] uses 2D Gaussians as in 2DGS, with multi-view depth and normal regularization to improve surface quality, while RaDe-GS [17] enables depth and normal rasterization for 3D Gaussians to support similar regularization. Other works introduced more explicit primitives, including planes [18, 19], optimizable geometry through learnable opacity maps [20], and soup of planes for dynamic reconstruction [21]. While these methods excel at representing flat surfaces with clean geometry, they typically sacrifice rendering quality and struggle to model phenomena that are better explained by volumetric effects, rather than surfaces. Some methods enforce planar constraints only as regularization losses, such as Guo et al. [22] that uses Manhattan world assumptions on semantically segmented regions and Chen et al. [23] that enforces plane normal consistency in textureless regions. Although helpful, regularizers can be difficult to tune. Our approach instead explicitly detects and optimizes planes within scene reconstruction, avoiding such issues.

**3D plane detection and reconstruction**   Another research direction *detects* planar surfaces in an initial 3D reconstruction and fits planes only to detected regions, extending single-image plane detection [24, 25] to multi-view settings. PlanarNeRF [26] adds a plane-predicting MLP branch to NeRF, supervised via ground truth labels or plane detection consistency across frames, but prevents plane MLP gradients from affecting the geometry prediction branch. PlanarRecon [8] reconstructs a sparse feature volume, which is decoded into plane features and clustered. AirPlanes [9] and NeuralPlane [27] build 3D-consistent plane embeddings per 3D point, emphasizing semantic priors for accurate detection. While we also use semantic knowledge, our method jointly detects and optimizes planes alongside scene reconstruction, allowing geometry to benefit from planar constraints. Further, unlike these methods, our approach yields full scene reconstructions suitable for novel view synthesis, vs. a coarse surface reconstruction.

**Hybrid representations**   Recent hybrid 2D-3D approaches have explored planar surface representation. Kim and Lim [28] integrate meshes into 3DGS for indoor scenes, using SAM [29] to detect planar surfaces and represent them with meshes while employing 3D Gaussians for other objects. Zanjani et al. [30] combine SAM segmentation with normal estimation to lift 2D plane descriptors to 3D, clustering the planar Gaussians using a tree structure. In contrast, our method offers a simpler solution by representing the scene with a mixture of 2D and 3D Gaussians. This design remains fully compatible with the 3DGS rendering pipeline, eliminating the need for complex hybrid mesh handling, or hierarchical tree structures.

## 3   Method

Given $N$ posed images $\{I_c\}$ and $M$ planar surfaces $\{P_p\}$, each specified by binary image masks $\{\mathcal{M}_{p,c}\}$, we aim to reconstruct a *hybrid* novel view synthesis method that combines a classical 3DGS model with a 2D piecewise planar representation of the scene. Our goal is to reconstruct the scene so that the planar surfaces are accurately recovered and compactly represented by 2D Gaussian primitives, while the rest of the scene is modeled with 3D Gaussians, with the key objective of avoiding the *artifacts* that can typically be seen when using 3D primitives to model planar surfaces; see Figure 1.

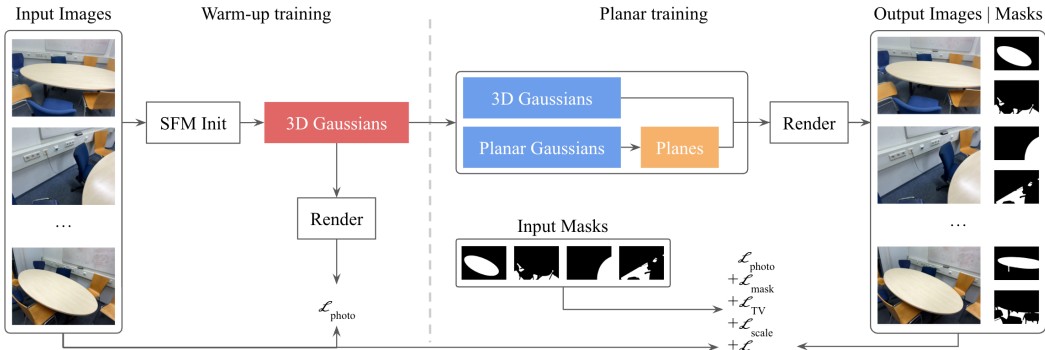

Figure 2: **Overview** – Training of our model is split into two parts: warm-up, in which 3D Gaussians are trained as in [6] using a photometric loss; and planar training, in which 3D Gaussians and planar Gaussians are trained along with the parameters of the planes to which planar Gaussians are locked. Planar training is performed in alternating phases, with Gaussian parameters frozen while plane parameters are optimized, and vice versa. Legend: learnable (warm up) , learnable (Gaussian phase) , learnable (plane phase).

### 3.1 Hybrid representation

Our representation consists of $M$ planes $\mathcal{P}=\{P_p\}$, each characterized by its 3D origin and normal $(\mathbf{o}_p, \mathbf{n}_p)$. The geometry of each each plane $P_p$ is represented through a set of 2D Gaussians $\mathcal{G}=\{\mathbf{g}_k\}_{k=1}^{K_k}$ such that,

$$\mathbf{g}_k = \mathcal{N}(\mu_k, \Sigma_k), \quad \mu_k \in P_k, \quad \Sigma_k \in \mathbb{R}^{2\times2}. \tag{1}$$

Here, $\mu_k$ is the center of the $k$-th Gaussian on the plane $P_p$, and $\Sigma_k$ is the 2D covariance matrix, parametrized with a 2D *in-plane* rotation $\mathbf{R}_k$ and a 2D diagonal scale matrix $\mathbf{S}_k$. The plane-to-world transformation matrix is defined as $\mathbf{T}_{\text{pw}}=\text{hom}(\mathbf{R}, \mathbf{o})$, where $\mathbf{R}$ is any rotation matrix that satisfies $\hat{z}=\mathbf{R}\mathbf{n}$ with $\hat{z}$ being the unit vector along the z-axis in the world frame. Thus, the degrees of freedom of planar Gaussians can be mapped to world coordinates through the rigid transformation:

$$\bar{\mu}_k = \mathbf{T}_{\text{pw}}[\mu_k; 0; 1], \quad \bar{\Sigma}_k = \mathbf{T}_{\text{pw}}\,\text{diag}(\Sigma_k, 1, 1)\,\mathbf{T}_{\text{pw}}^\top \tag{2}$$

yielding a standard 3D Gaussian primitive representation suitable for rendering. The remaining scene geometry is represented by unconstrained 3D Gaussians $\bar{\mathcal{G}}=\{\bar{\mathbf{g}}_k\}_{k=1}^{\bar{K}}$:

$$\bar{\mathbf{g}}_k = \mathcal{N}(\bar{\mu}_k, \bar{\Sigma}_k), \quad \mu_k \in \mathbb{R}^3, \quad \Sigma_k \in \mathbb{R}^{3\times3} \tag{3}$$

All Gaussians have view-dependent colors $\mathbf{c}$ represented as Spherical Harmonics, and opacity $\alpha$ as in vanilla 3DGS. To reconstruct the scene with our hybrid representation, we need to optimize the degrees of freedom of planes $\mathcal{P}$, 2D planar Gaussians $\mathcal{G}$, and 3D freeform Gaussians $\bar{\mathcal{G}}$. We begin our optimization with a warm-up stage using only 3D Gaussians (for N=3500 iterations). After that, we begin our planar reconstruction where in each round of optimization we: (i) dynamically initialize plane parameters by *robustly* fitting planes to the current representation (section 3.2); (ii) alternate between optimizing plane and Gaussian parameters (section 3.2); (iii) densify our representation through a (slightly modified) MCMC densification, due to the challenges of optimizing compact-support functions (section 3.4).

### 3.2 Plane initialization

For compactness of notation, let us drop our indices, and consider the binary mask $\mathcal{M} \leftarrow \mathcal{M}_{c,p}$ for the $p$-th planar surface in the $c$-th view, and denote with $\pi$ the function that projects a 3D point to the coordinate frame of the $n$-th image. We start by selecting all the Gaussians (i) whose mean projects into the mask, (ii) that are sufficiently opaque, and (iii) that lie within a shell of the expected ray termination of the $n$-th image:

$$\tilde{\mathcal{G}} = \{\bar{\mathbf{g}}_k \mid \pi(\bar{\mu}_k)\in\mathcal{M},\ \alpha_k>\kappa,\ |D(\pi(\bar{\mu}_k)) - d_k|<\delta\}, \tag{4}$$

where the thresholds $\alpha_{\text{th}}=0.1$ and $d_{\text{th}}=0.05$ are hyper-parameters that control this selection process, and where $D$ is the expected ray termination map (i.e. depth map), and $d_k$ is the depth of the

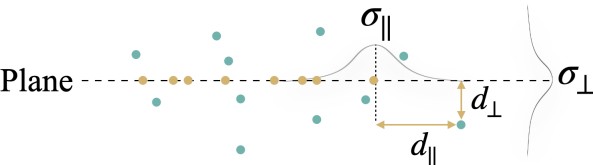

Figure 3: **Planar Relocation** – A freeform Gaussian (teal) gets relocated to the plane to become a planar Gaussian (brown), when both its distace to the plane ($d_\perp$) and along ($d_\parallel$) the plane are small.

Gaussians. We then extract a candidate plane $P$ by RANSAC optimization on the point cloud that samples the Gaussians:

$$P, \mathcal{I} = \text{RANSAC}\big(\{x \sim \bar{\mathbf{g}} \mid \bar{\mathbf{g}} \in \tilde{\mathcal{G}}\}, \epsilon\big) \tag{5}$$

where we accept $P$ as a viable plane candidate only whenever the mean inlier residual is lower than $\epsilon$. The set $\mathcal{I}$ includes the indexes of Gaussians in $\tilde{\mathcal{G}}$ that are inliers of the RANSAC process. We further discard planes that are too small with set $\mathcal{I}$ having a smaller size than 100. Once a plane corresponding to $\mathcal{M}$ has been accepted, all the semantic masks for that plane $p$ are excluded from subsequent RANSAC runs. The plane initialization process is repeated for remaining masks, after each completed round of plane and Gaussian optimization, as described in Section 3.3.

**Snapping** We then remove the discovered inliers from the set of 3D Gaussians $\bar{\mathcal{G}} \leftarrow \bar{\mathcal{G}} \setminus \mathcal{I}$, and add them to our set of 2D Gaussians $\mathcal{G} \leftarrow \mathcal{G} \cup \mathcal{I}$. During the latter operation, we clip 3D Gaussians to 2D to become planar by transforming to the local plane coordinates, and set the third component of their means and scales to zero. Further, only rotation about the z-axis in local plane coordinates is preserved

**Active set update** If the accepted plane $\mathcal{P}_i$ has an angular distance below a threshold to an already existing plane, while its origin $\mathbf{o}_i$ also has a small Euclidean distance to the closest Gaussian center on that plane, we merge the two planes. Otherwise, the plane is added as a new plane to the active set of planes $\mathcal{P}$. In merging, we assign the new plane's Gaussians to the previously found one. This allows our optimization to merge planar areas that have only been partially observed in any view.

## 3.3 Optimization

We optimize our representation by block-coordinate descent, starting each round of optimization by only optimizing the plane parameters for a fixed number of 10 iterations, and then freezing these, and optimizing the Gaussian parameters (both 2D and 3D) for another 100 iterations. This alternation in optimization is critical to avoid instability; see an ablation in figure 7. In the first optimization block, within each iteration, the parameters of the $p$-th plane within the $c$-th image are optimized by the loss:

$$\underset{\mathbf{o}_p, \mathbf{n}_p}{\arg\min} = \underbrace{\|I_c - \tilde{I}_c\|_1}_{\mathcal{L}_{\text{photo}}} + \lambda_{\text{mask}} \underbrace{\|\mathcal{M}_{c,p} - \tilde{\mathcal{M}}_{c,p}\|_1}_{\mathcal{L}_{\text{mask}}}, \tag{6}$$

where $\tilde{\mathcal{M}}$ is the predicted plane mask, obtained by rendering the mixture of Gaussians with binarized color (white for planar, and black for 3D), and alpha blended using the original Gaussian opacities during the rasterization. In the second optimization block, we optimize all Gaussian parameters jointly:

$$\underset{\mathcal{G}, \bar{\mathcal{G}}}{\arg\min} \quad \mathcal{L}_{\text{photo}} + \lambda_{\text{mask}} \mathcal{L}_{\text{mask}} + \lambda_{\text{TV}} \mathcal{L}_{\text{TV}} + \lambda_{\text{scale}} \mathcal{L}_{\text{scale}} + \lambda_{\text{opacity}} \mathcal{L}_{\text{opacity}}, \tag{7}$$

where $\mathcal{L}_{\text{TV}}$ is the total depth variation regularization from Niemeyer et al. [10], $\mathcal{L}_{\text{scale}}$ is the scale regularizer and $\mathcal{L}_{\text{opacity}}$ is the opacity regularizer from Kheradmand et al. [13] that vanishes the size of Gaussians that are unconstrained by the photometric loss. Note that planar Gaussians move *rigidly* during plane optimization (6), and move locally in the plane during Gaussian optimization (7), as only their 2D in-plane parameters are optimized.

## 3.4 Planar relocation

We follow 3DGS-MCMC [13] in our training dynamics. For densification of planes, we rely on relocating low-opacity Gaussians to locations of dense high-opacity Gaussians, as this allows

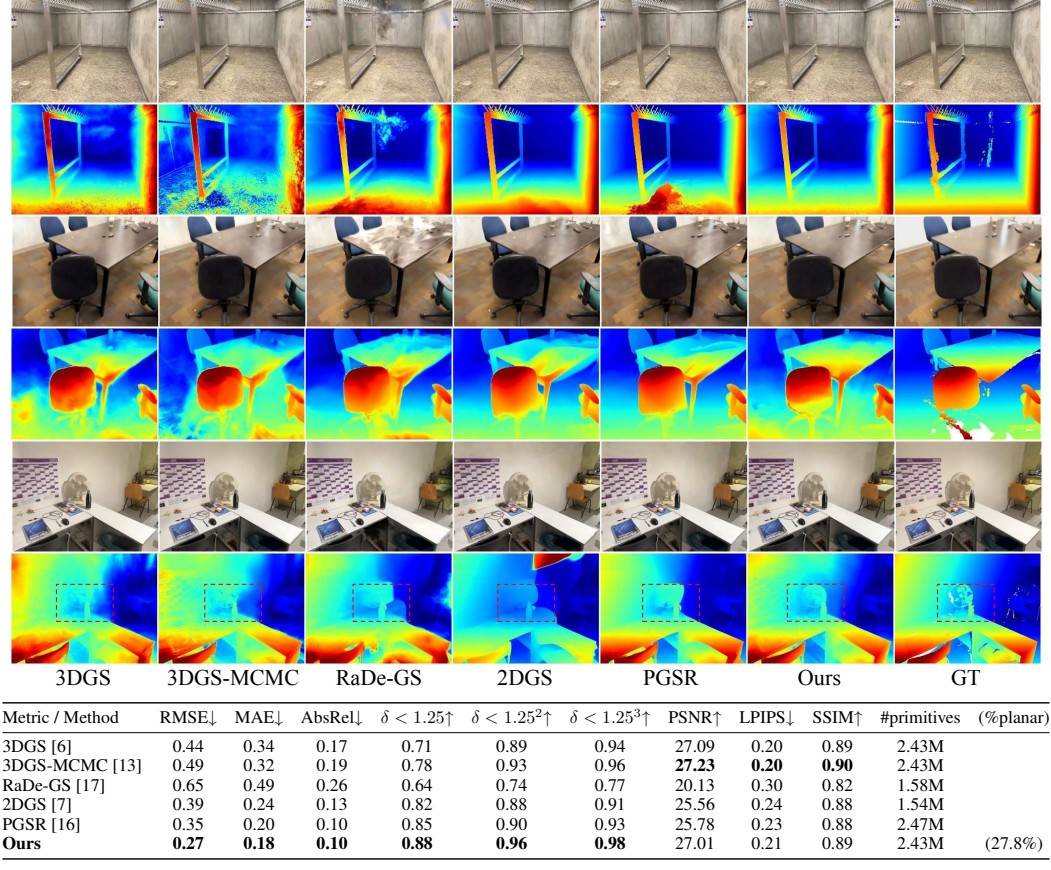

| | 3DGS | 3DGS-MCMC | RaDe-GS | 2DGS | PGSR | Ours | GT |

| Metric / Method | RMSE↓ | MAE↓ | AbsRel↓ | $\delta < 1.25$↑ | $\delta < 1.25^2$↑ | $\delta < 1.25^3$↑ | PSNR↑ | LPIPS↓ | SSIM↑ | #primitives | (%planar) |
|---|---|---|---|---|---|---|---|---|---|---|---|
| 3DGS [6] | 0.44 | 0.34 | 0.17 | 0.71 | 0.89 | 0.94 | 27.09 | 0.20 | 0.89 | 2.43M | |
| 3DGS-MCMC [13] | 0.49 | 0.32 | 0.19 | 0.78 | 0.93 | 0.96 | **27.23** | **0.20** | **0.90** | 2.43M | |
| RaDe-GS [17] | 0.65 | 0.49 | 0.26 | 0.64 | 0.74 | 0.77 | 20.13 | 0.30 | 0.82 | 1.58M | |
| 2DGS [7] | 0.39 | 0.24 | 0.13 | 0.82 | 0.88 | 0.91 | 25.56 | 0.24 | 0.88 | 1.54M | |
| PGSR [16] | 0.35 | 0.20 | 0.10 | 0.85 | 0.90 | 0.93 | 25.78 | 0.23 | 0.88 | 2.47M | |
| **Ours** | **0.27** | **0.18** | **0.10** | **0.88** | **0.96** | **0.98** | 27.01 | 0.21 | 0.89 | 2.43M | (27.8%) |

Figure 4: **Novel View Synthesis** – Quantitative and qualitative results show significant improvement in predicted depth compared to previous work, while maintaining comparable rendering quality to the full 3D representations.

transferring between 3D and 2D/planar Gaussians. However, the number of Gaussians on planes, especially when the plane has weak texture, is usually low, leading to a slow densification rate for planes / planar Gaussians. To address this issue, whenever a freeform Gaussian projects into the current mask $\pi(\bar{\mu}_k) \in \mathcal{M}$, and it is *sufficiently close* to the currently reconstruction, we stochastically re-locate it to the plane. To measure distance, we identify the 2D Gaussian with the smallest Euclidean distance to $\bar{\mu}_k$, and measure its distance in the direction of the plane normal $d_\perp$, and the one along the plane $d_\parallel$; see Figure 3. We stochastically relocate this if both distances are sufficiently small, as expressed by the following Bernoulli distribution:

$$p \sim \mathcal{B}(\beta), \; \beta = \left[ 1 - \Phi \left( \frac{d_\perp}{\sigma_\perp} \right) \right] \cdot \left[ 1 - \Phi \left( \frac{d_\parallel}{\sigma_\parallel} \right) \right], \tag{8}$$

where $\Phi$ is the cumulative distribution function of a Gaussian, and $\sigma_\perp$ and $\sigma_\parallel$ are hyper-parameters that control the stochastic re-location.

## 4 Results

We validate our proposed method for scene reconstruction through the novel view synthesis task on common indoor scene datasets, assessing both rendered image and depth quality metrics (section 4.1). We then show an application of our method to mesh extraction for planar surfaces (section 4.2). Finally, we validate our design choices through an ablation study on different aspects of the method (section 4.3). We provide our implementation details in the supplementary material.

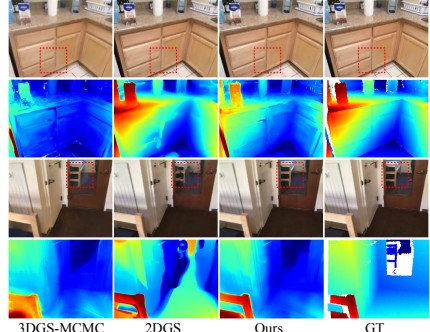

| Metric | 3DGS-MCMC | 2DGS | Ours |
|---|---|---|---|
| RMSE↓ | 0.46 | 0.60 | **0.40** |
| MAE↓ | 0.37 | 0.44 | **0.31** |
| AbsRel↓ | 0.19 | 0.23 | **0.16** |
| $\delta < 1.25 \uparrow$ | 0.61 | 0.63 | **0.70** |
| $\delta < 1.25^2 \uparrow$ | 0.87 | 0.77 | **0.90** |
| $\delta < 1.25^3 \uparrow$ | 0.95 | 0.83 | **0.97** |
| PSNR↑ | 20.18 | 21.44 | **21.75** |
| LPIPS↓ | 0.29 | 0.30 | **0.27** |
| SSIM↑ | 0.83 | 0.85 | **0.86** |
| # primitives | 500K | 809K | 500K |
| (% planar) | | | (17.6%) |

Figure 5: **Novel View Synthesis on ScanNetv2** – Our method outperforms baselines in image and depth quality on ScanNetv2 despite sparse camera views.

### 4.1 Novel View Synthesis – Figures 4 and 5

We evaluate our hybrid representation's novel view synthesis on common indoor scene reconstruction benchmarks and provide comparisons with both state-of-the-art fully 3D representations and 2D surface reconstruction approaches. We show a significant improvement in the reconstructed surface geometry while maintaining high visual quality.

**Datasets** We perform evaluations on common indoor scene benchmarks ScanNet++[31] and ScanNetv2[32], as they primarily feature indoor scenes with flat textureless surfaces suitable for the task at hand. ScanNet++ provides dense scenes with SfM camera poses and sparse point clouds, designed primarily for 3D reconstruction approaches that follow the NeRF [5] paradigm. Conversely, the legacy version of ScanNet i.e. ScanNetv2 offers sparser views without SfM information. Our method works with or without initial sparse point clouds, enabling reconstruction initialized with sparse SfM point cloud on ScanNet++ and experiments with randomly initialized point clouds on ScanNetv2. For ScanNet++, we use 11 training scenes with ground truth meshes for depth derivation, utilizing iPhone video streams, sampling every 10th frame for training at 2× downsampling and every 8th for testing. We chose the scenes that are diverse in their content and contain various planar surfaces. For ScanNet, we evaluate on 5 scenes with sufficient overlapping views of planar surfaces following the data preparation scheme of [27]. The 2D plane masks were generated using PlaneRecNet [25] and propagated through the image sequence with SAMv2 video processor [29].

**Baselines** We compare against SOTA reconstruction methods, both fully 3D representations and 2D surface reconstruction methods. For 3D representations, we compare with vanilla 3DGS [6], and 3DGS-MCMC [13] as it is more robust version to random initializations, and has higher rendering quality. Within photometric surface reconstruction methods, we compare to 2DGS [7] as a widely used state-of-the-art, as well as to PGSR [16] and RaDe-GS [17], which more recently report improved depth quality. All methods are trained for 30K iterations.

**Metrics** We use the common image quality metrics PSNR, SSIM and LPIPS for evaluating the rendered RGB. Further, we choose depth as a strong indicator for the quality of the reconstructed surface geometry. We provide depth quality metrics by computing the rendered depth as the expected ray termination at each pixel. We report RMSE, MAE and average absolute error relative to ground truth depth (AbsRel). Additionally, we provide depth accuracy percentage at different error thresholds similar to [33]. The metrics are computed only on the defined portion of the ground-truth depths. We further report the total number of primitives in our model and the percentage that are planar (and thus can be represented more compactly).

**Analysis** Quantitative and qualitative results across both datasets show significant improvement in depth accuracy compared to *all* baselines. Notably, our method achieves comparable image quality to SOTA 3D representations on dense ScanNet++ scenes while surpassing them in depth quality, evidenced by sharper geometry reconstruction in qualitative examples. The slight PSNR difference with 3D methods reflects a trade-off: our constrained geometry enforces correct structure, while unconstrained methods can inflate PSNR by fitting view-dependent effects with incorrect geometry.

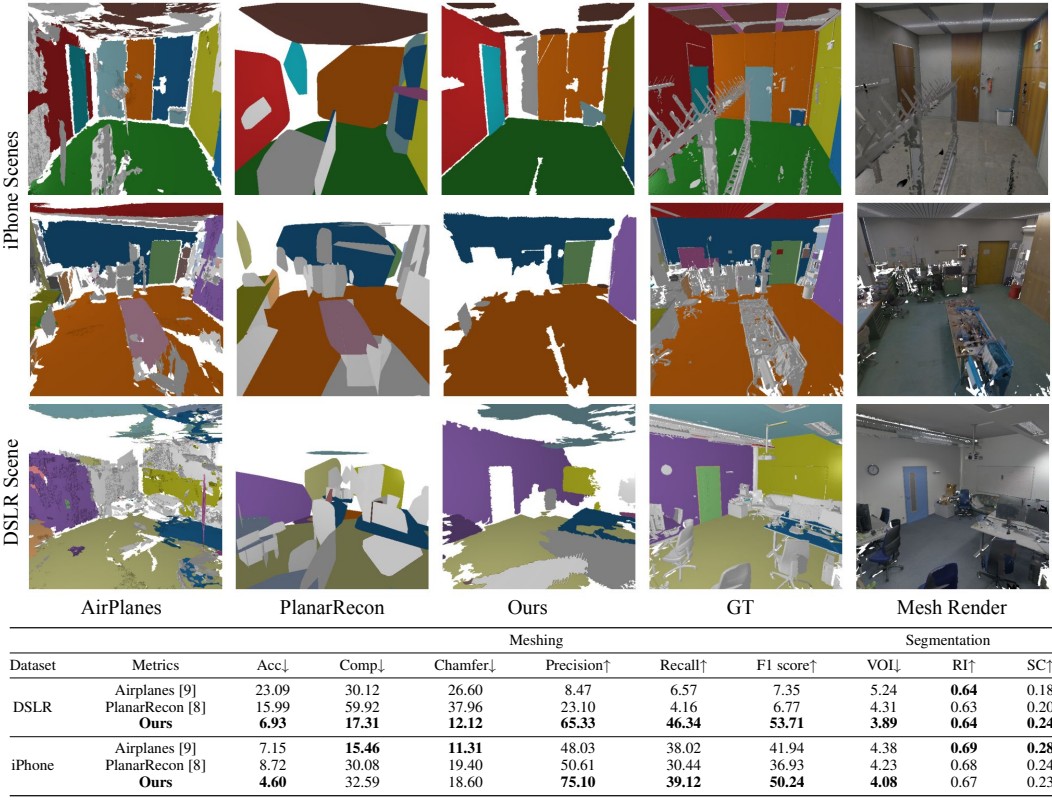

| Dataset | Metrics | Meshing | | | | | | Segmentation | | |
|---------|---------|------|-------|---------|-----------|--------|----------|------|-----|-----|
| | | Acc↓ | Comp↓ | Chamfer↓ | Precision↑ | Recall↑ | F1 score↑ | VOI↓ | RI↑ | SC↑ |
| DSLR | Airplanes [9] | 23.09 | 30.12 | 26.60 | 8.47 | 6.57 | 7.35 | 5.24 | **0.64** | 0.18 |
| | PlanarRecon [8] | 15.99 | 59.92 | 37.96 | 23.10 | 4.16 | 6.77 | 4.31 | 0.63 | 0.20 |
| | **Ours** | **6.93** | **17.31** | **12.12** | **65.33** | **46.34** | **53.71** | **3.89** | **0.64** | **0.24** |
| iPhone | Airplanes [9] | 7.15 | **15.46** | **11.31** | 48.03 | 38.02 | 41.94 | 4.38 | **0.69** | **0.28** |
| | PlanarRecon [8] | 8.72 | 30.08 | 19.40 | 50.61 | 30.44 | 36.93 | 4.23 | 0.68 | 0.24 |
| | **Ours** | **4.60** | 32.59 | 18.60 | **75.10** | **39.12** | **50.24** | **4.08** | 0.67 | 0.23 |

Figure 6: **Mesh Extraction** – Our method shows consistent results across iPhone and DSLR captures, while baselines typically overfit to one camera type. Qualitatively, our approach extracts complete meshes for most target planes with fewer inaccurate plane detections (shown in gray) compared to baselines. Target planes are shown with distinct colors on the ground truth.

In the sparser ScanNetv2 scenes, our approach delivers superior performance in both depth and image quality, leveraging the planar prior of indoor environments to overcome the geometric ambiguity that challenges pure 3D methods in sparse captures. Our method also substantially outperforms 2DGS in both image fidelity and depth accuracy metrics.

## 4.2 Mesh Extraction – figure 6

Our method enables mesh extraction from reconstructed 3D planar surfaces. For each plane, we un-project all 2D segmentation masks to 3D by computing ray-plane intersections, yielding a point cloud. This point cloud is downsampled using fixed-size voxels and rasterized onto plane coordinates to create an occupancy grid. We then use Marching Squares for contour extraction (We omit small contours with less than 100 points), followed by ear-clipping triangulation to produce the final mesh. We evaluate the quality of the retrieved mesh for the planar surfaces and compare our method to planar reconstruction methods.

**Datasets** We use ScanNet++ to extract planar surface meshes. We show results both on the subset of this dataset captured by iPhone and also the DSLR subset, showing that our method can handle different camera models, while previous methods usually overfit to one modality. For ground truth, we follow the approach of Watson et al. [9] to obtain a ground truth planar mesh. We then only consider the subset of planes in the ground truth mesh that we have annotated segmentation masks for each scene. We provide details on selecting these planes in Appendix E.

**Baselines** We compare against previous planar reconstruction methods AirPlanes [9] and PlanarRecon [8] that provide extracted planar mesh as output of their methods. We follow the same evaluation setting as in the original papers on the iPhone subset of the dataset. For DSLR images, we crop the images to the specified FoV in each baseline to match their training distribution.

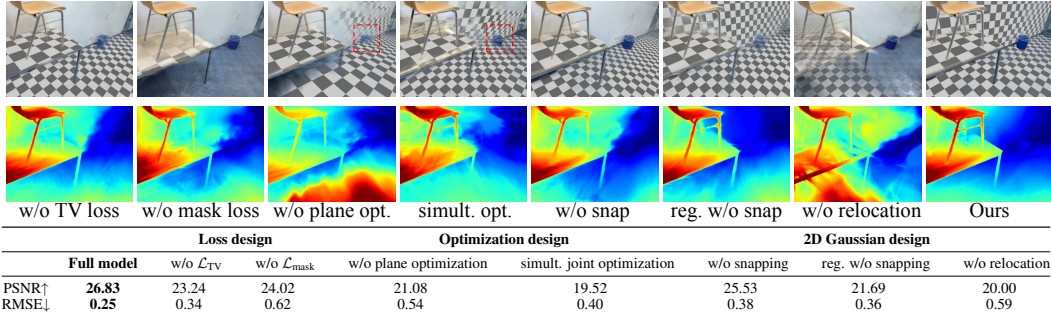

| | Loss design | | | Optimization design | | | 2D Gaussian design | |
|---|---|---|---|---|---|---|---|---|
| | Full model | w/o $\mathcal{L}_{TV}$ | w/o $\mathcal{L}_{mask}$ | w/o plane optimization | simult. joint optimization | w/o snapping | reg. w/o snapping | w/o relocation |
| PSNR↑ | **26.83** | 23.24 | 24.02 | 21.08 | 19.52 | 25.53 | 21.69 | 20.00 |
| RMSE↓ | **0.25** | 0.34 | 0.62 | 0.54 | 0.40 | 0.38 | 0.36 | 0.59 |

Figure 7: **Ablation on design choices** – Loss components and optimization strategy are critical, with simultaneous plane-Gaussian optimization causing significant drops. 2D Gaussian snapping greatly improves depth accuracy compared to regularization alternatives. Similarly, Gaussian relocation is essential.

**Metrics**   We report mesh accuracy metrics including accuracy, precision, recall, completeness and Chamfer distance as defined in Ye et al. [27]. We also provide mesh segmentation metrics that evaluate how well detected plane segments match ground truth segments following [9].

**Analysis**   Our method outperforms the baselines on DSLR images subset of the dataset. Unlike previous methods that are trained on specific modalities (i.e. phone camera) and struggle to transfer to different camera models (i.e. DSLR camera), our approach maintains consistent mesh quality due to having zero-shot mesh extraction on test scenes through photometric reconstruction. Additionally, our method outperforms PlanarRecon on iPhone data, while having competitive performance to AirPlanes. Qualitative results reveal that both PlanarRecon and AirPlanes extract extraneous planes with numerous random small fragments, resulting in unsightly and impractical meshes. In contrast, our method produces clean planar surfaces, yielding a more coherent and usable reconstruction.

### 4.3  Ablation – Figure 7

We ablate our design choices and additionally test our method's robustness to random point cloud initialization (in table 1).

**Loss design**   We ablate the effect of $\mathcal{L}_{mask}$ and $\mathcal{L}_{TV}$. Although removing these losses reduces the image quality by some margin, it affects depth quality more significantly. Qualitative rendering shows that $\mathcal{L}_{mask}$ contributes significantly to detecting and growing 2D Gaussians.

**Optimization design**   Our method is based on optimizing Gaussians and plane parameters together in an alternating fashion. We show that fixing plane parameters with no optimization degrades our results both quantitatively and qualitatively. Simultaneous joint optimization of Gaussians and planes also affects the results negatively. In Figure 7, note how the floor plane gets stuck above the ground level, as revealed by its intersection with the bin.

**2D Gaussian design**   Using hybrid 2D/3D Gaussians is one of the main components of our design. Therefore, we ablate the necessity of having 2D Gaussians by disabling snapping as described in Section 3.2. This shows a significant drop in depth accuracy, which is also evident in qualitative results. As an alternative to snapping, we can regularize the smallest scale component in planar Gaussians. However, we find that this approach is difficult to tune and provides suboptimal results. Finally, we ablate our densification process with relocation of Gaussians to planes. Without relocation, planes are not fully detected, with the planar Gaussians comprising the plane maintaining low opacity. Furthermore, some of the Gaussians remain close to the plane while not being detected as belonging to that plane.

# 5   Conclusions

We introduce 3D Gaussian Flats, a hybrid 2D/3D Gaussian representation that accurately models planar surfaces without sacrificing rendering quality. Our method jointly optimizes 2D Gaussians constrained to planar surfaces alongside free-form Gaussians for the remaining scene. By leveraging semantic segmentation masks, we predict both a full 3D representation and semantically distinct planes for planar mesh extraction in indoor scenes. Our approach achieves state-of-the-art depth estimation on indoor scene benchmarks while maintaining high image quality. Additionally, our planar mesh extraction method generalizes across different camera models, overcoming domain gap limitations that typically cause previous methods to fail.

**Limitations**   Our reliance on initial 3DGS reconstruction often generates insufficient Gaussians in flat areas with no texture, although this potentially can be addressed via more adaptive densification strategies. Further, using a weak spherical harmonics appearance model still leads to building extra geometry to compensate for view-dependent effects, which a stronger appearance model would resolve. Additionally, we depend on 2D semantic masks from SAMv2 that may contain errors, but our method will naturally improve alongside advances in semantic segmentation. Finally, our RANSAC-based approach, while robust, introduces computational overhead that extends training time. We believe our hybrid representation opens exciting new avenues for research into more efficient approaches that balance geometric precision with visual fidelity.

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

# A    Full mesh extraction results – Figures 8 to 10

We evaluate our hybrid representation on the task of full mesh extraction using the method from [7], we do it in addition to the planar-only mesh extraction experiments presented in Section 4.2, concatenating the two meshes together and comparing them to common benchmarks from Section 4.1.

**Datasets**    We evaluate on ScanNet++ [31], a common indoor scene benchmark, as well as on subset of suitable indoor/outdoor scenes from ETH3D [34], which provides high quality mesh, and is more challenging because of sparse image supervision.

**Baselines**    For ScanNet++ we reuse the models trained on iPhone data stream and evaluated on the task of NVS in Section 4.1 to access mesh quality reconstruction. On ETH3D, in addition, we evaluate Gaussian Opacity Fields (GOF) [35], an extention of 2DGS for higher quality mesh reconstruction, and DNSplatter [36], a method supervising 3DGS with mono-depth [2]

To obtain the mesh, we use TSDF fusion with the median depth estimate for 3DGS, 2DGS, DNSplatter and ours, rather than the expected ray termination as in default settings (i.e., average depth). For PGSR we use their proposed unbiased depth computation, and for Gaussian Opacity Fields we extract the mesh using the level set of the Gaussians, hence the mesh is not colored.

**Metrics**    We use the same metrics as for meshing task in planar mesh experiments Section 4.2. We compute the F1-score at 5 cm threshold. For both of the datasets, we use every 8th image as a test image.

**Analysis**    We provide full mesh renders along with the metrics on ScanNet++ in Figure 8. For ETH3D, in addition to mesh renders in Figure 10, we provide rendered novel views from the test set in Figure 9. Note that captured planar surfaces are unbiased and outline well the structures of the scenes. Moreover, on in the sparse view setting on ETH3D dataset we achieve a notable rendering quality improvement.

# B    Additional ablations – Tables 1 and 2

**Random initialization**    We analyze the effect of having sparse point cloud initialization versus random initialization in our method on 11 DSLR scenes from ScanNet++ [31]. for random initialization we do 5000 iterations in our warmup stage, as opposed to the usual 3500. We show that our method maintains the robustness to random initialization similar to 3DGS-MCMC [13], and despite a drop in number of planar Gaussians, it achieves comparable depth and image quality metrics to our method when initialized with SfM sparse point cloud.

Table 1: **Ablation on initialization –** Our method is robust to random initialization and achieves comparable performance to when initialized with SfM point cloud.

| Method | PSNR↑ | SSIM↑ | LPIPS↓ | RMSE↓ | MAE↓ | AbsRel↓ | #primitives | (%planar) |
|---|---|---|---|---|---|---|---|---|
| 3DGS-MCMC (SfM) | 23.38 | **0.87** | **0.24** | 0.41 | 0.24 | 0.26 | 1.13M | |
| **Ours (SfM)** | **23.42** | **0.87** | **0.24** | **0.20** | **0.13** | **0.12** | 1.13M | (31%) |
| **Ours (Random)** | 23.30 | 0.86 | 0.25 | 0.21 | 0.14 | 0.13 | 1.13M | (21%) |

**Full metrics set for ablation on design choices**    We provide the full set of metrics for ablation on design choices (described in section 4.3) in the table 2.

---

[2] Note that the released codebase for DNSplatter does not support multiple camera models (different camera intrinsics) for aligning mono-depth to SfM points, therefore we cannot easily report the metrics for 'Electro' and 'Terrace' scenes.

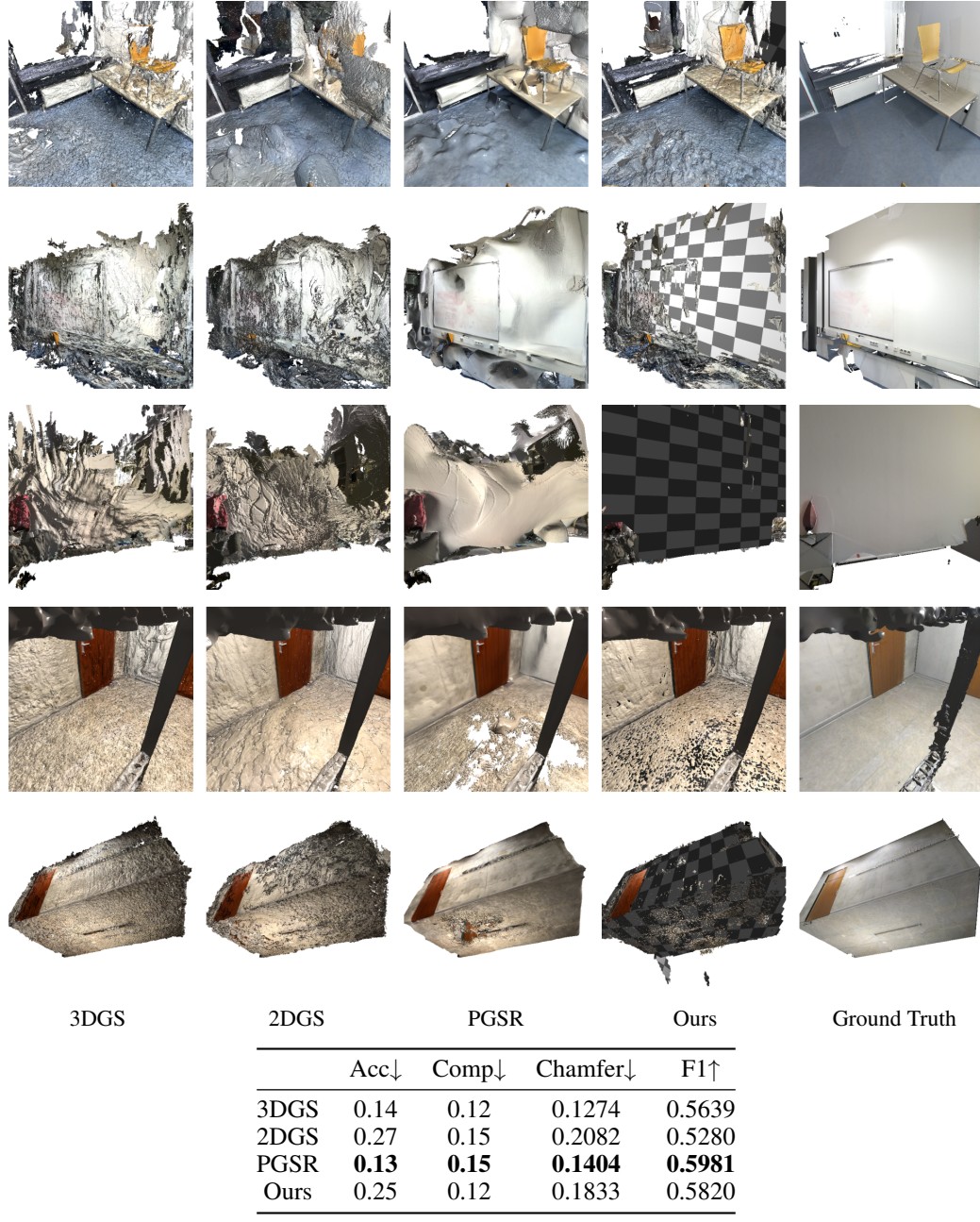

|       | Acc↓ | Comp↓ | Chamfer↓ | F1↑ |
|-------|------|-------|----------|------|
| 3DGS  | 0.14 | 0.12  | 0.1274   | 0.5639 |
| 2DGS  | 0.27 | 0.15  | 0.2082   | 0.5280 |
| PGSR  | **0.13** | **0.15** | **0.1404** | **0.5981** |
| Ours  | 0.25 | 0.12  | 0.1833   | 0.5820 |

Figure 8:   **Full Mesh Extraction Results on ScanNet++** – Out method achieves competitive performance for surface reconstruction, while mainatining the rendering quality. Checkered surfaces indicate different planes, planes are usually behind the TSDF-extracted mesh as they represent unbiased surfaces. Some of the meshes are shown from outside of the indoor scene to highlight the planar alignment.

## C   Additional video and 3D mesh results

We provide video renderings of RGB and depth for our method compared to baselines in https://theialab.github.io/3dgs-flats. Video results best capture the significant enhancement of our approach over baselines in depth estimation and accurately modeling scene geometry.

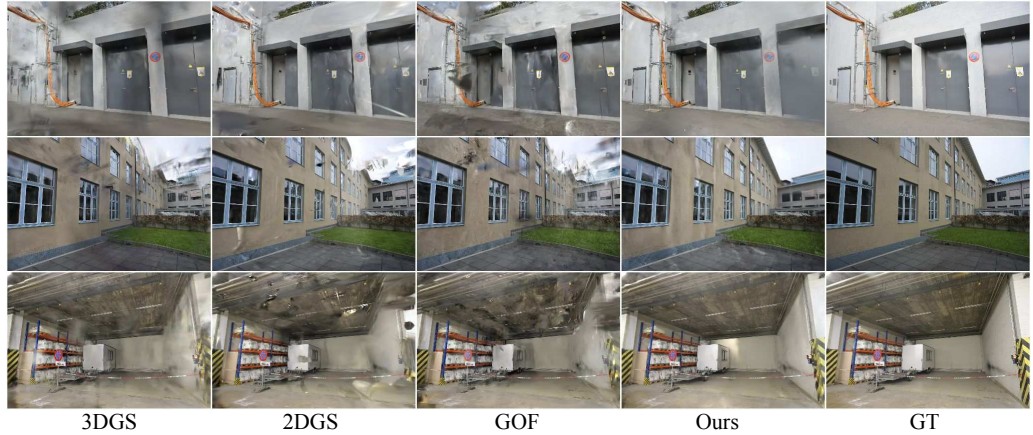

| | 3DGS | 2DGS | GOF | Ours | GT |

| Method | Electro | | | Terrace | | | Delivery area | | |
|---|---|---|---|---|---|---|---|---|---|
| | PSNR↑ | LPIPS↓ | SSIM↑ | PSNR↑ | LPIPS↓ | SSIM↑ | PSNR↑ | LPIPS↓ | SSIM↑ |
| 3DGS | 16.45 | 0.38 | 0.72 | 20.77 | 0.27 | 0.78 | 19.48 | 0.29 | 0.83 |
| 2DGS | 16.40 | 0.41 | 0.72 | 20.82 | 0.29 | 0.79 | 19.26 | 0.35 | 0.81 |
| GOF | 17.34 | 0.36 | 0.71 | 20.80 | 0.27 | 0.75 | 19.40 | 0.33 | 0.79 |
| PGSR | – | – | – | – | – | – | 16.64 | 0.41 | 0.69 |
| DNSplatter | – | – | – | – | – | – | 19.56 | 0.24 | 0.77 |
| Ours | **18.72** | **0.31** | **0.75** | **22.57** | **0.22** | **0.81** | **22.56** | **0.21** | **0.87** |

Figure 9: **Rendering Results on ETH3D Scenes** – Our method outperforms the baselines in terms of rendering quality on this set of sparse view outdoor/indoor scenes, and the planar representation is crucial for achieving good novel view synthesis in sparse scenarios.

Table 2: **Ablation on design choices** – Loss components and optimization strategy are critical, with simultaneous plane-Gaussian optimization causing significant drops. 2D Gaussian snapping greatly improves depth accuracy compared to regularization alternatives. Similarly, Gaussian relocation is essential.

| | PSNR↑ | LPIPS↓ | SSIM↑ | RMSE↓ | MAE↓ | AbsRel↓ |
|---|---|---|---|---|---|---|
| Full model | **26.83** | **0.27** | **0.86** | **0.25** | **0.18** | **0.09** |
| **Loss design:** | | | | | | |
| w/o $\mathcal{L}_{TV}$ | 23.24 | 0.34 | 0.82 | 0.34 | 0.24 | 0.13 |
| w/o $\mathcal{L}_{mask}$ | 24.02 | 0.32 | 0.83 | 0.62 | 0.53 | 0.29 |
| **Optimization design:** | | | | | | |
| w/o plane optimization | 21.08 | 0.37 | 0.80 | 0.54 | 0.43 | 0.24 |
| simult. joint optimization | 19.52 | 0.38 | 0.79 | 0.40 | 0.32 | 0.18 |
| **2D Gaussian design:** | | | | | | |
| w/o snapping | 25.53 | 0.31 | 0.84 | 0.38 | 0.31 | 0.17 |
| reg. w/o snapping | 21.69 | 0.35 | 0.81 | 0.36 | 0.28 | 0.15 |
| w/o relocation | 20.00 | 0.37 | 0.80 | 0.59 | 0.50 | 0.28 |

# D    Additional qualitative results – Figures 11 and 12

We provide more qualitative evidence for the performance of our method compared to 2DGS [7], 3DGS [6] and 3DGS-MCMC [13] baselines on the ScanNet++ [31] dataset in figure 11. The results show how baselines particularly struggle with reconstructing accurate geometry for the textureless areas while our method significantly improves upon these methods in depth estimation and keeps the visual quality of images.

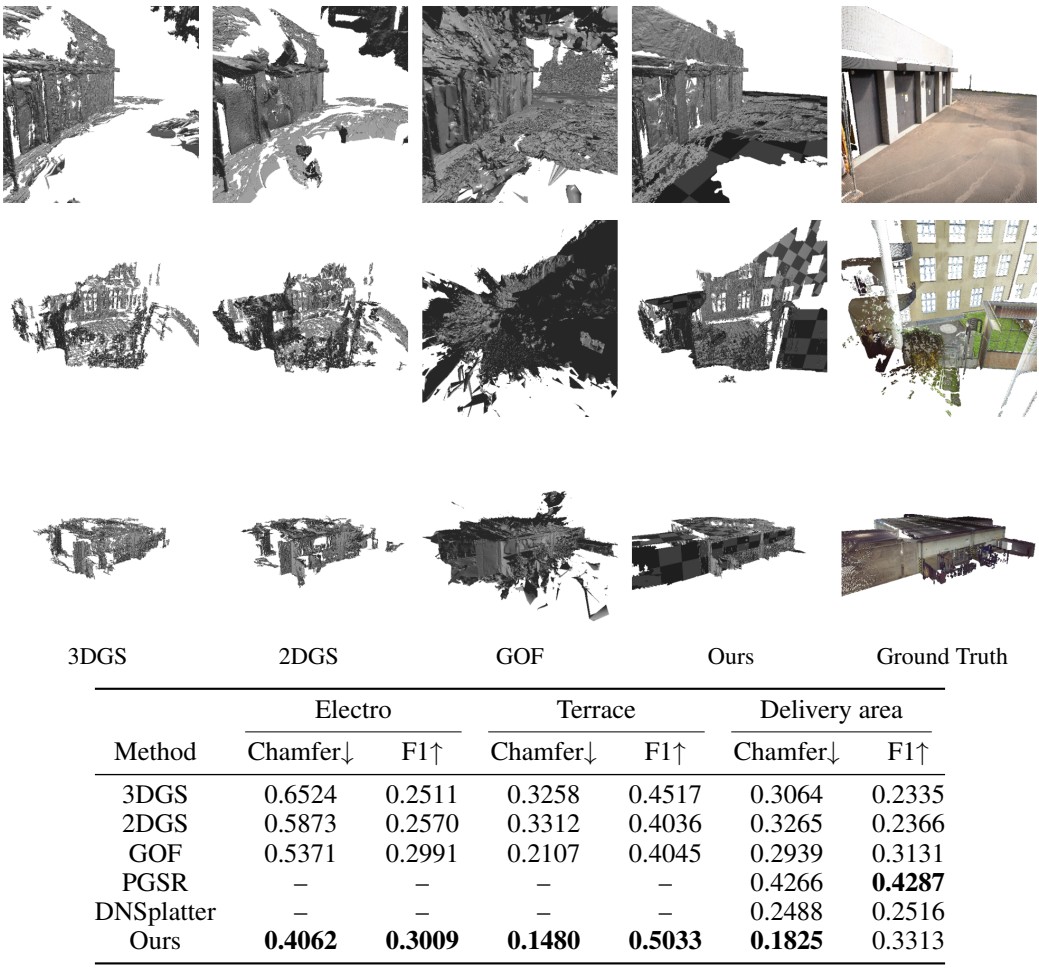

| 3DGS | 2DGS | GOF | Ours | Ground Truth |

| Method | Electro | | Terrace | | Delivery area | |
| --- | --- | --- | --- | --- | --- | --- |
| | Chamfer↓ | F1↑ | Chamfer↓ | F1↑ | Chamfer↓ | F1↑ |
| 3DGS | 0.6524 | 0.2511 | 0.3258 | 0.4517 | 0.3064 | 0.2335 |
| 2DGS | 0.5873 | 0.2570 | 0.3312 | 0.4036 | 0.3265 | 0.2366 |
| GOF | 0.5371 | 0.2991 | 0.2107 | 0.4045 | 0.2939 | 0.3131 |
| PGSR | – | – | – | – | 0.4266 | **0.4287** |
| DNSplatter | – | – | – | – | 0.2488 | 0.2516 |
| Ours | **0.4062** | **0.3009** | **0.1480** | **0.5033** | **0.1825** | 0.3313 |

Figure 10: **Full Mesh Extraction Results on ETH3D Scenes** – Our method outperforms the baselines.

Further, we provide more visualization for our estimated planes on ScanNet++ [31] dataset, showcasing the perfect alignment of our planes with the detected planar surfaces in figure 12.

# E    Input planar masks

**2D semantic masks**    Our method relies on input consistent 2D segmentation masks of planar surfaces. To obtain these masks, we can either annotate each image collection manually or automate the process for larger scenes. To automatically obtain the 2D segmentation masks, we employ PlaneRecNet [25] and SAMv2 video segmentation model [29], to create an annotation pipeline. We first input images to PlaneRecNet to obtain 2D plane annotations that are not semantically consistent across the image collection. We set the plane probability threshold to 0.5. While this method works well on iPhone images, it produces fewer plane annotations for DSLR images, that are out of distribution for its network trained on iPhone data. We then input these unmatched masks as seed to SAMv2. In order to do that, we order image collections that are not already sampled from a video. We propagate masks from the initial frame in 16-frame chunks of the sequence to the next 15 frames, and match SAMv2's prediction with any subsequent 2D masks output from [25], using Hungarian matching with an IoU metric. Although largely effective, this process is prone to error accumulation through mask propagation. However, we assume resultant masks are semantically consistent across the image sequence. We provide sample segmentation of an input sequence in the supplementary video and on the website.

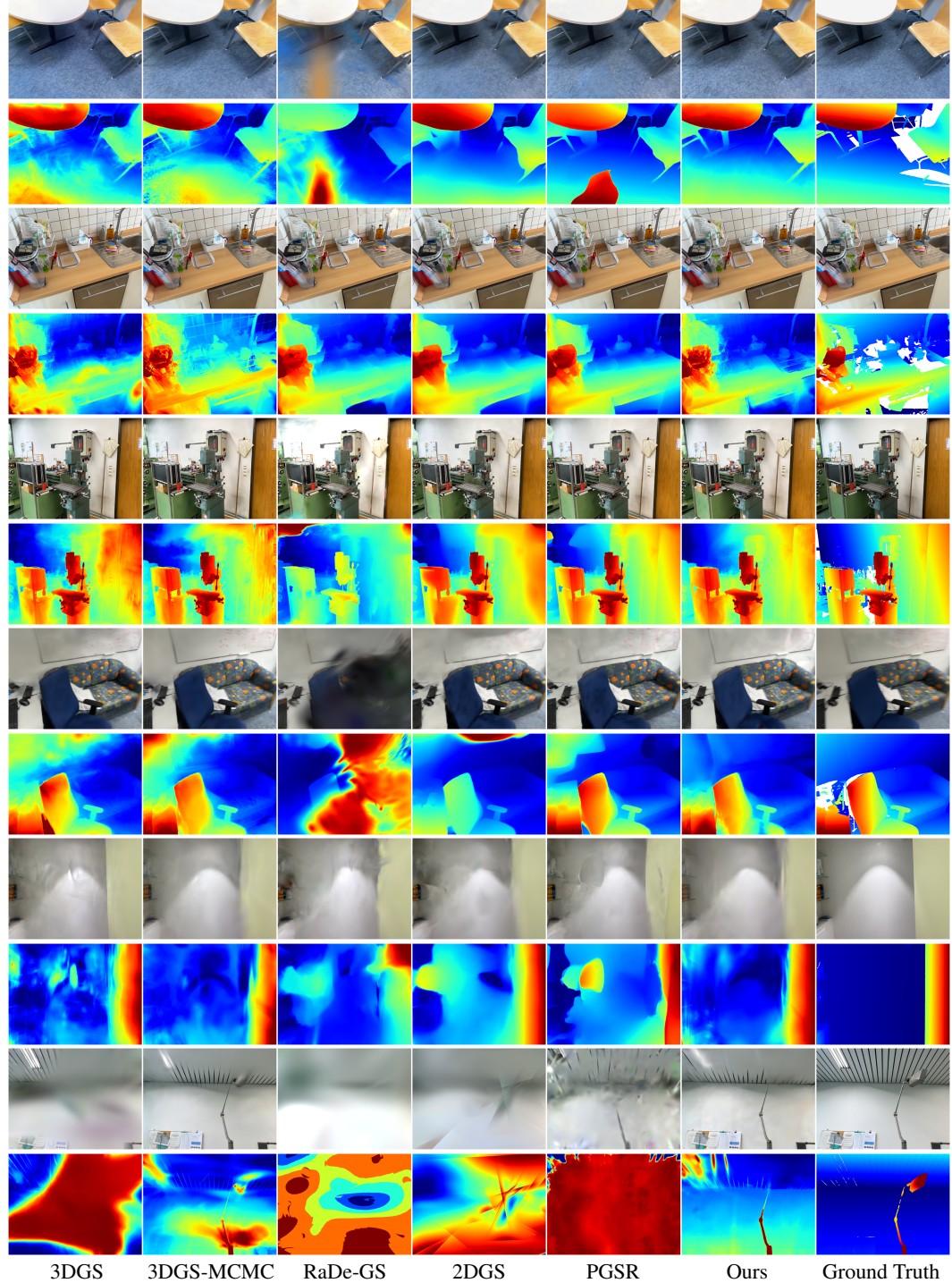

| 3DGS | 3DGS-MCMC | RaDe-GS | 2DGS | PGSR | Ours | Ground Truth |

Figure 11: **Novel view synthesis and depth –** Qualitative results on ScanNet++ iPhone dataset show our superior performance in both image quality and depth estimation in novel views. Note the limitation of the quality of Gaussian Splatting based methods for textureless surfaces, which makes the plane fitting procedure less robust.

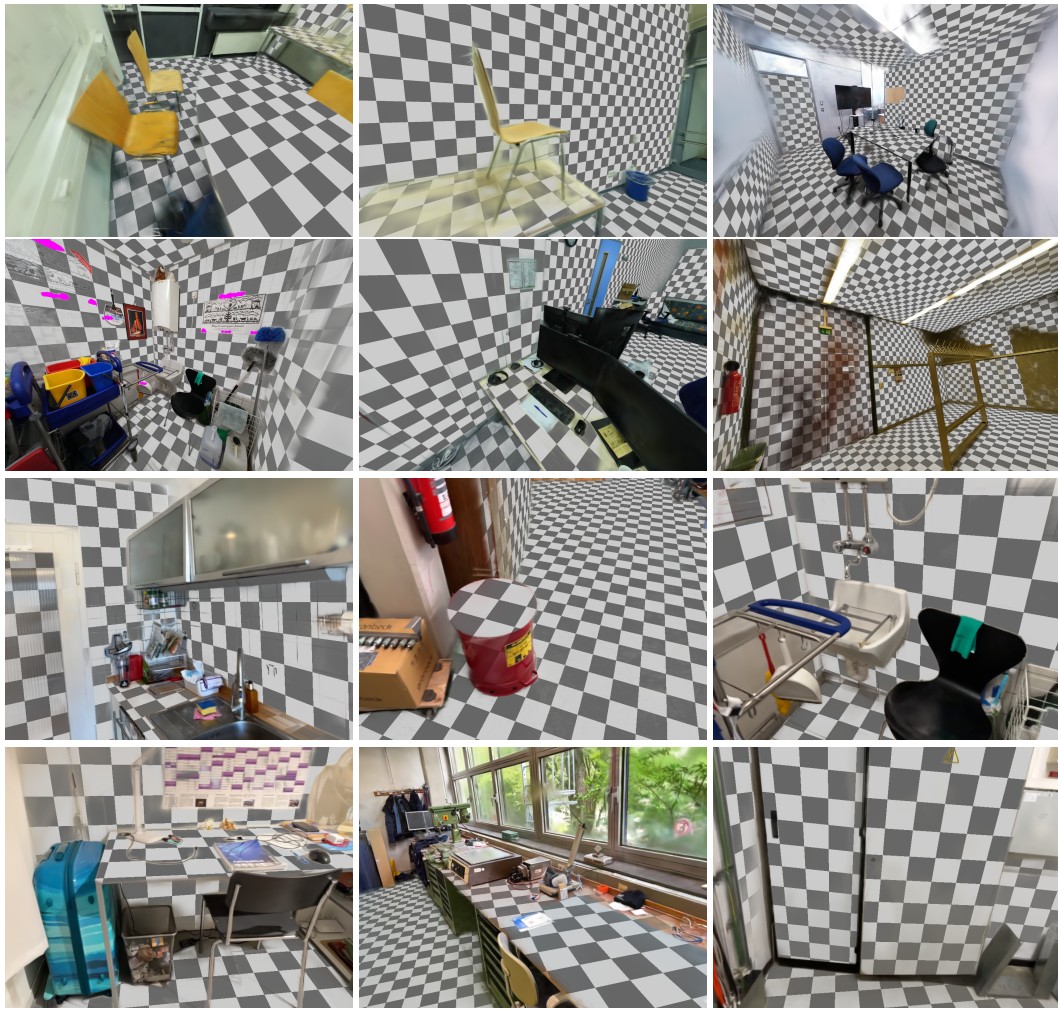

Figure 12: We provide visualizations of our output planes on the rendered test views of ScanNet++ DSLR streams (top 2 rows) and iPhone stream (bottom 2 rows). Pink markings are due to the anonymization of the original ScanNet++ dataset. While some planar surfaces are missed due to lack of manual 2D planar mask annotation, the captured planes are reconstructed faithfully.

**Masked ground truth meshes**  For the planar mesh extraction task, we only consider planes with annotated segmentation masks from the ground truth mesh, as the 2D plane segmentation task is orthogonal to our method. To identify the relevant subset of planes, we unproject points from the ground truth depth maps that correspond to each plane according to its segmentation mask. We then fit a plane to each resulting point cloud using RANSAC and compile these fitted planes into a set $S$. We match planes from the ground truth mesh to those in set S by applying two criteria: the normal cosine distance must be less than 0.99 to at least one plane in $S$, and the distance between their closest points must be less than 0.1. Doing this allows for computational efficiency and increased robustness to missing or undersegmented planes in the input 2D annotations.

**Code**  We release our code[3] publicly for reproducibility purposes and to facilitate future research in this area. We base our code on the 3DGS-MCMC paper [13] and additionally use SAMv2 [29], and PlaneRecNet [25] to generate masks. The baselines are evaluated using their official released code [7, 6, 16, 17, 13, 8, 9]. We further utilize AirPlanes [9] code to compute meshing metrics.

---

[3]https://github.com/theialab/3dgs-flats

# F  Hyperparameters settings

We use $\sigma_\perp$ and $\sigma_\parallel$ as hyper-parameters that control the stochastic re-location. These parameters are chosen depending on the metric scale of the dataset, and are defined in millimeters. For both datasets we used $\sigma_\perp = 0.01$ and $\sigma_\parallel = 0.3$. We observe that setting $\lambda_{mask} = 0.1$, yields best results empirically. For regularizers, we use $\lambda_{TV}$=0.1, $\lambda_{scale}$=0.01 and $\lambda_{opacity} = 0.01$ following [10] and [13]. We use the same scheduling policy for learning plane origin and normal (rotation) as for the Gaussian means the vanilla 3DGS. All experiments were conducted on a single A6000 ADA GPU, with 46GB memory. The method runs for approximately 1 hour for a single ScanNet++/ScanNetV2 scene, which is comparable to PGSR [16], the second best method for geometric quality according to our experiments and 1.5× longer than 3DGS-MCMC [13], the best method for Novel View Synthesis. The training time is increased due to the RANSAC overhead and block-coordinate descent optimization of planar parameters. Additionally, mesh extraction takes $\sim 3$ minutes and SAM mask propagation is on average 7 minutes long, depending on the scene type. We believe that the training time can be reduced in future work with addition of customized CUDA kernels.

