# OpenReview forum: "3D Gaussian Flats: Hybrid 2D/3D Photometric Scene Reconstruction"
_NeurIPS.cc/2025/Conference — NeurIPS 2025 poster_

### Official Review · Reviewer_wJbn · 2025-06-01

**Clarity:** 2
**Significance:** 2
**Originality:** 3
**Rating:** 4
**Confidence:** 4

**Summary:**

This paper proposes a novel method that combines 3D Gaussians and 2D Gaussians binded on the detected planar to represent non-planar and planar areas, respectively. To optimize the hybrid representation, the authors propose a plane initialization method and multi-stage optimization process, which demonstrates helpful in the ablation study. The experiment results indicate the effectiveness of this paper to some extent.

**Questions:**

Concerns: See Weakness part.

Typos:

1. line 131, wrong ref for section 3.2
2. line 140, no $d_{th}$

**Ethical Concerns:**

["NO or VERY MINOR ethics concerns only"]

**Final Justification:**

Having reviewed all comments, the authors have addressed most of my concerns. Thus, I raise my rating to BA.

**Limitations:**

yes

**Quality:**

2

**Strengths And Weaknesses:**

**Strengths:**

1. This paper propose a hybrid representation to leverage the geometric prior of detected planars in the indoor scenes, which improves the quality of depth map estimation.
2. The experiment results demonstrate this hybrid representation surpasses previous planar reconstruction methods on diverse datasets.

**Weakness:**

1. The major weakness of this paper is the confused application areas described in the introduction and experiment settings. For novel view synthesis, this method shows performance drop compared with 3DGS or 3DGS-MCMC. For surface reconstruction, this method can only reconstruct planar areas, with large missing regions for other areas (as shown in Fig. 6). Thus I concern whether comparing this method with 3DGS(NVS) or PGSR(Surface Rec) is necessary and fair. Fig 4 and Fig 5 only demonstrate this method can obtain better depth estimation results.
2. For depth estimation, the comparison with 2DGS is also not very fair. The hybrid representation leverages planar prior to refine the depth map of planar regions, while 2DGS relys on no planar consumption and can generalize to ourdoor scenes. The comparison with 2DGS + planar regularizer (such as structnerf) is necessary to demonstrate the effectiveness of the hybrid representation, although the authors say "regularizers can be difficult to tune", which is not a straightforward conclusion to me.
3. Some statements about the relationship between 2D Gaussians, 3D Gaussians and planar are confused in Sec 3. Is each 2D Gaussians binded extractly on the plane for its definition in Eq.(1) and (2)? If so, what's the smallest Euclidean distance of 2D Gaussians in line 182? Should it be always zero? If Sec 3.4 describes how to transform a 3D gaussian near the plane to a 2D Gaussian binded on the plane, I highly recommend do not use "freeform" or "planar" and change the subsection name, cause desifincation has specific meanings in 3D GS.
4. In line 147, may M correspond to several planar masks? Because the author say subsequent RANSAC runs after detecting one valid plane.
5. For experiments, in line 246, the authors mention this method generalize to different camera models well, while previous fails. The reason is unclear. This method relys heavily on the performance of PlaneRecNet. Is this robustness caused by the diverse training data of PlaneRecNet, which contains different types of camera models? Why previous methods fail?

---

> ### Author Rebuttal · Authors · 2025-07-31
>
> We thank the reviewer for recognizing the improved depth and planar mesh quality outputs of our method compared to baselines.
>
> **Confus[ing] application areas.**
> Our method is motivated by the goal of jointly improving photometric and geometric accuracy in indoor scenes, which often feature large, *weakly textured* planar surfaces. Methods such as 3DGS and 2DGS, which rely solely on photometric optimization without explicit geometric priors, struggle to recover accurate geometry in such regions, particularly under specular or low-texture conditions. For example, see Figure 3 in DN-Splatter, which concludes that `The Gaussian based methods Splatfacto, SuGaR, and 2DGS are trained on only photometric losses and thus severely struggle to capture the scene geometry in low texture environments`.
>
> **Q: Performance drop in novel view synthesis (NVS).**
> As shown in Figures 4 and 5, our method achieves slightly lower RGB rendering quality than 3DGS-MCMC but outperforms 3DGS, while significantly improving depth accuracy. Depth serves as a strong indicator of geometric quality.
>
> This leads to a key question: does poor geometry affect NVS performance? The answer is yes. Inaccurate geometry can degrade rendering from out-of-distribution viewpoints noticeably (those not well-covered by training views). We show this in our supplementary video. This issue is well-documented in the literature [3, 4], which highlights how weak geometry can lead to rendering artifacts under more aggressive viewpoint shifts from training views, than is usually present in NVS test sets of common datasets.
>
> **Q: Only reconstructs planar areas for surface reconstruction.**
> While we explicitly apply geometric priors only to planar regions, we can still extract full-scene meshes using the same strategy as 2DGS. As shown in the `Incomplete Mesh Extraction Evaluation` section (see Response to Reviewer `bJrA`), overall our method produces more accurate meshes than all baselines, demonstrating the benefit of incorporating planar priors. Figure 6 presents planar-only mesh results to *allow for a fair comparison* with prior plane-based reconstruction methods, which typically report metrics solely for planar regions.
>
> **Unfair comparison to 2DGS.**
> The reviewer raises an important point: incorporating planar priors does significantly improve geometric accuracy, especially in depth estimation. This is indeed a core contribution of our method: accurately detecting and fitting 3D planes, which vanilla 2DGS lacks and thus struggles in planar regions.
>
> Importantly, our hybrid design is not simply combining 2D and 3D Gaussians everywhere. We use 2D Gaussians exclusively in the detected 3D planar regions where geometry is known, and reserve 3D Gaussians for non-planar areas without priors, where their higher expressiveness helps photometric reconstruction. This separation is deliberate: 2D Gaussians are compact and accurate when structure is known, while 3D Gaussians handle unconstrained regions. Further, while we implemented our solution on top of 3DGS (i.e. non-planar splats are 3D gaussians), there is no reason for which what we proposed could be implemented on top of a 2DGS backbone (i.e. non-planar splats would be 2D gaussians).
>
> As for regularization-based methods like StructNeRF, these are not directly applicable, as with the advent of 3DGS we moved NVS training workloads away from ray-based implementations and towards tile rasterizers. Further, while StructNeRF imposes the planar consistency loss within any superpixel, our technique only activates the planar optimization if a likely planar structure has been detected by RANSAC. While a direct comparison is not possible due to the ray vs. tile implementation, considering the annotations as “candidates” allows for better handling of false positives.
> We will clarify this in the paper.
>
> **Generalization to different camera models.**
> Our method is not uniquely dependent on PlaneRecNet. PlaneRecNet is used as a more automatic prompt generation alternative for SAMv2 compared to manual prompting. SAMv2 performs the actual segmentation. While PlaneRecNet is limited by its training on ScanNetV2 to one camera model, our generalization comes from the robustness of SAMv2. In fact, the results on ETH3D in this rebuttal use manual prompts via a user-click UI to SAMv2, showing strong performance even without PlaneRecNet.
>
> **Answer to Q3 (Method details).** Each 2D Gaussian is bound exactly to a detected plane, as defined in Eqs. (1) and (2). In line 182, we compute two distances: the residual distance from a 3D Gaussian to the plane (z-axis), and the in-plane (xy) distance to the nearest 2D Gaussian. This helps decide whether to reassign nearby 3D Gaussians without growing the plane indefinitely. We agree that terms like “freeform” and “planar” could be confusing and will revise the subsection title accordingly.
>
> **Answer to Q4 (Method details).** Yes, M corresponds to several planar masks for a particular 3D plane from multiple views. The 2D masks are propagated from one view or multiple user clicks using the SAMv2 video model.
>
> **Bibliography**
>
> - [3] Warburg, Frederik, et al. "Nerfbusters: Removing ghostly artifacts from casually captured nerfs." Proceedings of the IEEE/CVF International Conference on Computer Vision. 2023.
> - [4] Goli, Lily, et al. "Bayes' rays: Uncertainty quantification for neural radiance fields." Proceedings of the IEEE/CVF Conference on Computer Vision and Pattern Recognition. 2024.

---

> > ### Comment · Reviewer_wJbn · 2025-08-06
> > **Further discussion**
> >
> > Thank you for the authors' response.
> > Having reviewed all comments, particularly Reviewer bJrA's valuable insights, I now better understand this work's application scope. Unlike prior methods (e.g., 2D-GS, GOF, PGSR, or 3D-GS), this paper specifically targets indoor scenes characterized by textureless surfaces and planar regions that pose challenges to 3D reconstruction. The proposed novel representation aims to jointly address novel view synthesis and surface reconstruction. Given its specialized focus and stronger scene assumptions, the method should demonstrate superior performance in both rendering quality and geometric accuracy. Consequently, quantitative comparisons with surface reconstruction methods (complete mesh, and please do not ignore PGSR, a stronger baseline) on indoor scenes are essential. While the rebuttal provides supplementary experimental results, Reviewer bJrA notes that omitted implementation details significantly impact evaluation metrics, and thus the research community's assessment. These methodological clarifications should have been included in the original submission. I therefore maintain my BJ rating.

---

> > > ### Author Response · Authors · 2025-08-06
> > > **Response to reviewer **wJbn**:**
> > >
> > > We thank the reviewer for their valuable comment. We're glad that the motivation of our work is more clear now. We agree that extended evaluations done in the rebuttal period have added value for assessing our method, and we aim to incorporate these results in the final version of our paper.
> > >
> > > - **Regarding reviewer bJrA comment on evaluation metrics, the official code online for the baselines does not reproduce the results reported by Reviewer bJrA (we invite you to verify on your own)**, despite extensive hyperparameter tuning and evaluating under different depth estimation strategies (e.g., median vs. expected termination). We rely on official released code bases of baselines and have an evaluation pipeline following the reviewer's instruction (i.e. training and mesh extraction script taken from 2DGS). We would sincerely welcome further clarification from the reviewer regarding their experimental setup.
> > > We have further reached out to a third-party outside the authors team to run independent evaluation on the baselines and verify our numbers, we will consequently update on their results.
> > >
> > > - We are committed to incorporating any clarifying details of our evaluation pipeline raised during the rebuttal period in the paper. Further, **all of our code and evaluation scripts will be made publicly available** to ensure full transparency and enable reproducibility by the community.
> > >
> > > - As noted in the paper, balancing high-quality rendering with accurate geometry is a challenging task, and many methods sacrifice one for the other. Our approach improves **both** simultaneously, as shown in Figure 4, with gains in image quality and rendered depth accuracy that directly reflect better geometry. We also include partial mesh comparisons to plane-based methods, which is standard in plane-based methods and targets regions affected by our prior. We agree that full mesh comparisons would further strengthen the paper and we are committed to adding these results in the final version of the paper.  We are working on providing results for comparison to PGSR.

---

> > > > ### Author Response · Authors · 2025-08-07
> > > > **Answer to wJbn**
> > > >
> > > > We would once again like to thank the reviewer for taking the time to consider our rebuttal and the accompanying discussion. Following our exchange with reviewer ‘bJrA’, we believe a consensus has been reached regarding the reported metrics. To further support this, we engaged an independent expert (external to the author team) to reproduce the baseline evaluations. Their results closely match our originally reported numbers (0.3209 vs 0.3131 F1@5cm on `Delivery area’).
> > > >
> > > > In light of this additional verification and the constructive dialogue with reviewer ‘bJrA’, we respectfully ask the reviewer to re-evaluate the discussion, taking into account the latest comments and validations.
> > > > We further provide comparison on full mesh extraction task with PGSR as reported below:
> > > >
> > > > PGSR results for ETH3D
> > > >
> > > > `Delivery area` (indoor)
> > > > |                         | PSNR↑ | LPIPS↓ | SSIM↑ | Chamfer↓ | F1↑    |
> > > > |-------------------------|-------|--------|-------|----------|--------|
> > > > | PGSR (unbiased depth)   | 16.64 | 0.41   | 0.69  | 0.4266   | 0.4287 |
> > > > | Ours (mean depth)       | 22.56 | 0.21   | 0.87  | 0.2305   | 0.3281 |
> > > > | Ours (median depth)     | 22.56 | 0.21   | 0.87  | 0.1825   | 0.3313 |
> > > >
> > > >
> > > > PGSR depth is unbiased, hence very precise, however the mesh is incomplete in the regions of interest (where no texture or reflections are present).
> > > >
> > > >
> > > > Updated ScanNet++ results (with median depth and PGSR added)
> > > >
> > > > `ScanNet++ (11 scenes)`
> > > >
> > > > |              | PSNR↑ | LPIPS↓ | SSIM↑ | Acc↓  | Comp↓ | Chamfer↓ | F1↑    |
> > > > |--------------|-------|--------|-------|-------|-------|----------|--------|
> > > > | 3DGS         | 27.09 | 0.20   | 0.89  | 0.14  | 0.12  | 0.1274   | 0.5639 |
> > > > | 2DGS         | 25.56 | 0.24   | 0.88  | 0.27  | 0.15  | 0.2082   | 0.5280 |
> > > > | PGSR         | 25.78 | 0.23   | 0.88  | 0.13  | 0.15  | 0.1404   | 0.5981 |
> > > > | Ours         | 27.01 | 0.21   | 0.89  | 0.25  | 0.12  | 0.1833   | 0.5820 |
> > > >
> > > >
> > > > 2DGS is worse than 3DGS with median depth: it looks like depth maps fusion fails on flat textures where surfels do not align with the true surface, so these regions are missing.

---

### Official Review · Reviewer_BGYx · 2025-06-25

**Clarity:** 3
**Significance:** 3
**Originality:** 3
**Rating:** 5
**Confidence:** 4

**Summary:**

This paper presents 3D Gaussian Flats, a hybrid novel view synthesis approach that proposes using both 3D gaussians and planar-constrained 2D gaussians. This allows it to come close to 3D Gaussian Splatting's rendering quality while improving upon its depth reconstruction in flat areas (ie: avoiding "holes" in apparent surfaces) via planar constraints. The method requires image masks for each plane instance (obtained via SAMv2) in addition to camera images/poses as in standard 3DGS. The optimization begins with standard 3D gaussian optimization as in 3DGS before initializing planar surfaces. The planar surfaces are constructed by projecting 3D gaussians onto the image masks, and then running RANSAC based on the gaussian means. 3D Gaussians within a certain range to the plane are then relocated to planar-aligned 2D gaussians. Optimization then alternates between gaussian optimization and the optimization of planar parameters.

The approach is evalauted on indoor scenes from ScanNet++ and ScanNetv2 against common baselines such as 3DGS, MCMC, and 2DGS. It outperforms all baselines with respect to depth metrics, and is competitve with regards to appearance metrics such as PSNR. Mesh extraction is also evaluated, and the approach outperforms the Airplanes and PlanarRecon baselines.

**Questions:**

I provisionally feel positive about the paper. The main thing I'd be interested in seeing are even qualitative results on outdoor scenes such as in AV to see if road reconstruction looks better than in 3DGS and/or Tanks and Temples.

**Ethical Concerns:**

["NO or VERY MINOR ethics concerns only"]

**Final Justification:**

Thank you for answering my questions. I will maintain my positive rating, although I agree with other reviewers that the experimental section could be strengthened and would strongly recommend adding more baselines to the camera-ready.

**Limitations:**

Yes

**Quality:**

3

**Strengths And Weaknesses:**

Overall the paper writing is clear (although the figures could be enlarged/improved - especially the teaser where I don't think it's clear what "faux render" means) and the hybrid approach seems sensible. The results on ScanNet look compelling and the ablations are reasonable.

The main weakness I see is its reliance on accurate planar segmentation masks and its potential brittleness when the masks are not totally accurate. Furthermore, the evaluation section limits itself to ScanNet scenes whereas a wider evaluation would be interesting (Fig 1 shows a MipNeRF 360 scene but MipNeRF is not evaluated quantitatively). Even if dense ground truth depth is hard to find for most datasets, getting PSNR metrics on M360 for example would give us a sense of how much the hybrid approach degrades visual quality. And even if outdoor scene reconstruction is not a focus of this paper, it would be interesting to evaluate things like road reconstruction on AV datasets like Waymo or KITTI (which also contain sparse depth that could be used for evaluation), or on Tanks and Temples which comes with (slightly imperfect) laser-scan derived meshes.

---

> ### Author Rebuttal · Authors · 2025-07-31
>
> Thank you for recognizing our contribution in designing the hybrid 2D/3D representation and its ability to produce compelling results on the ScanNet dataset. We will improve the image quality and figure annotations in the camera-ready version.
>
> **Evaluation on outdoor scenes.**
> We focus on indoor scenes, where large, weakly textured planar surfaces pose challenges for 2DGS and 3DGS methods, making ScanNet a suitable benchmark. However, we agree that demonstrating generalization to other datasets including outdoor settings is valuable.
> As suggested by Reviewer `bJrA`, we evaluate our method on ETH3D dataset, which includes two outdoor scenes containing planar structures such as roads and building facades. These results show our method remains effective beyond indoor environments. Please refer to our response to Reviewer `bJrA` under `Insufficient Experimental Validation (i.e. our new experiments)` for details.
>
>
> **Reliance on 2D segmentation.**
> We recognize the reliance of our method on 2D segmentation as a limitation. However, thanks to recent progress in segmentation models (e.g., SAMv2), this requirement is becoming increasingly robust. These models keep the human in the loop through sparse but valuable point prompts, which come at low cost to the user and contribute to more reliable outputs.
>
>
> **Metrics for Mip-NeRF 360?**
> Please note that the metrics are already reported in the paper within the teaser.

---

> > ### Comment · Reviewer_BGYx · 2025-08-04
> >
> > Thank you for answering my questions. Re: the M360 metrics, are the PSNR numbers in Fig 1 for all of M360 or just the Garden scene? My request was for the former.

---

> > > ### Author Response · Authors · 2025-08-05
> > >
> > > Thanks for the clarification. The teaser only shows the quantitative result for this specific scene from MipNeRF360 dataset, which contains an interesting planar surface (i.e. the table, besides the ground). We chose this scene from MipNeRF360 because the specular part of the table in the center is commonly reconstructed inaccurately as a concave surface due to the ambiguity in explaining the specularity jointly with geometry and appearance.
> > > We appreciate the suggestion to include more scenes from this dataset in our evaluation. However, we note that most scenes in MipNeRF360 lack any planar surfaces apart from the ground, which limits their relevance to our specific analysis. The "Kitchen" and "Room" scenes do contain some planar surfaces, and we agree they could be of interest. However, they primarily depict indoor environments similar to those already included in our evaluation via ScanNet++ and ETH3D, and therefore may offer limited additional diversity.

---

### Official Review · Reviewer_T6jG · 2025-06-29

**Clarity:** 3
**Significance:** 2
**Originality:** 2
**Rating:** 4
**Confidence:** 3

**Summary:**

3D Gaussian Flats proposes a hybrid 2D/3D representation that jointly optimizes constrained planar (2D) Gaussians for modeling flat surfaces and freeform (3D) Gaussians for the rest of the scene. This end-to-end approach dynamically detects and refines planar regions within the scene. The core innovation lies in this adaptive, dual-representation strategy, enabling the creation of photorealistic digital twins that are robust to texture-less regions. The performance is supported by extensive experiments on ScanNetV2 and ablation studies.

**Questions:**

See "Strengths And Weaknesses" section.

**Ethical Concerns:**

["NO or VERY MINOR ethics concerns only"]

**Final Justification:**

Thanks for the author's response. My concerns have been largely addressed, but I'm keeping my current scores because of the visual quality.

**Limitations:**

See "Strengths And Weaknesses" section.

**Paper Formatting Concerns:**

The last page still have some margin left.

**Quality:**

3

**Strengths And Weaknesses:**

**Strengths**
1. It is interesting to segment planar objects from the scene and condition separate Gaussians to model them. This hybrid 2D/3D representation efficiently improve the reconstruction and view synthesis quality.
3. The paper is well-motivated and the structure is clean and easy to understand.

**Weaknesses** (including questions)
1. How does the method apply to outdoor scenes that also have planar parts? The author should include more experiments rather than only focusing on indoor scenes.
2. The qualitative comparison in Figure 1, 4, and 7 demonstrates marginal improvement in terms of visual quality. In addition, the figure seems quite low-res., potentially directly from sceenshots.
3. It is quite confusing that in Figure 6, 3D Gaussian Flats outperforms Airplanes in terms of “Acc” but inferior in “Comp.” And “Chamfer Distance”. The author should provide more explaination regarding this.

---

> ### Author Rebuttal · Authors · 2025-07-31
>
> Thank you for recognizing our novel hybrid representation and robust 3D plane fitting.
> We address your concerns below.
>
> **Outdoor scenes.**
> Our method is primarily motivated by scenes containing large, weakly textured planar surfaces, a common failure case for both 2DGS and 3DGS. Such surfaces are especially prevalent in indoor environments, which is why our main evaluation focuses on indoor datasets like ScanNet++.
> To demonstrate generalization to outdoor settings, we also evaluate two outdoor scenes from ETH3D, which contain roads and building facades with planar geometry. Additionally, our teaser shows the results on Mip-NeRF 360 outdoor scene with rendering quality metrics.
> These results show that our method remains effective beyond purely indoor environments. Please refer to our response to Reviewer `bJrA` under `Insufficient Experimental Validation (i.e. our new experiments)` for details.
>
>
> **Marginal improvement in visual quality.**
> We agree that the RGB renderings in Figures 1 and 4 show only modest visual differences a-la glance.
> However, these comparisons are best understood alongside their rendered depth maps, which reveal the underlying geometry.
>
>
> In both figures, methods without our planar prior tend to explain away specular highlights (e.g., the shiny table in Figure 1 and Figure 4) and low-texture regions (walls and floors in Figure 4) using volumetric effects instead of modeling the simple planar geometry paired with a more complex specular appearance or a flat texture.
>
>
> This incorrect geometry can noticeably degrade rendering quality when viewed from slightly out-of-distribution angles, beyond the well-covered test views, as shown in our supplementary video.
> As noted in `Line 230` of the paper, our approach strikes a better balance between geometric accuracy and photometric fidelity: we preserve correct planar structure due to our geometric prior while still capturing complex appearance effects such as specularity.
>
>
> We thank the reviewer for their point about figure resolutions, we will include figures at full resolution in the camera-ready version (note that we have full resolution results in the supplementary).
>
>
> **Mesh accuracy metrics.**
> Our method achieves higher accuracy because the predicted points lie closer to the ground truth surface. However, it has lower completeness since some planar regions (e.g., undetected or unlabeled planes) are missing. In contrast, the Airplanes method covers more surface area, but with less precision. Chamfer Distance reflects both of these aspects, so our conservative predictions which favors correctness over coverage leads to higher Chamfer distance on iPhone scenes.

---

### Official Review · Reviewer_bJrA · 2025-07-01

**Clarity:** 3
**Significance:** 2
**Originality:** 3
**Rating:** 4
**Confidence:** 4

**Summary:**

This paper introduces a novel method for representing 3D scenes by combining 2D and 3D Gaussians. Specifically, 2D Gaussians are employed for planar surfaces, while 3D Gaussians handle the remaining scene elements. The selection of 2D Gaussians is based on their projection onto pre-estimated plane masks and their adherence to a RANSAC-based plane fitting, allowing the method to automatically differentiate between 2D and 3D Gaussian assignments.

**Questions:**

- What is the mesh quality compared to 2DGS, Gaussian Opacity Fields and DNSplatter?
- How is the method different than simply having monodepth/mononormal supervision?
- What RANSAC did the authors use to get the plane parameters?
- What is the advantage of using a mix of 2D/3D Gaussians instead of only 2D ones?

I am willing to improve my score, but only if the authors show mesh quality comparisons on standard datasets against 2DGS, Gaussian Opacity Fields and DNSplatter.

**Ethical Concerns:**

["NO or VERY MINOR ethics concerns only"]

**Final Justification:**

The authors managed to address my concerns in their rebuttal. Thus, I improve my rating.

**Limitations:**

Yes

**Paper Formatting Concerns:**

No concerns

**Quality:**

3

**Strengths And Weaknesses:**

--- Strengths ---

The concept of integrating both 2D and 3D Gaussians within a single scene representation is interesting. However, the current experimental setup does not sufficiently motivate or justify the advantages of this hybrid approach.

--- Weaknesses ---

The paper in its current state shows several major weaknesses.

- Limited Justification for Hybrid Approach: The proposed method for identifying 2D Gaussians - relying on projections into pre-estimated plane masks and RANSAC for robustness - lacks a clear advantage over existing techniques. It is unclear how this approach surpasses methods that utilize monocular depth or normal supervision, such as DNSplatter, in terms of geometric accuracy. Furthermore, given that 2DGS has already demonstrated superior geometric accuracy with disc-based representations compared to 3D Gaussians, the rationale for a hybrid approach rather than exclusively using 2D discs is not well-motivated.
- Insufficient Experimental Validation: The experimental section is very weak, particularly concerning geometric accuracy comparisons. The paper primarily focuses on photometric/depth accuracy and some geometric comparisons with planar segmentation methods, neglecting crucial baselines. Essential comparisons with state-of-the-art 3D Gaussian Splatting (3DGS) methods that prioritize geometric accuracy (e.g., F1 score of the extracted mesh), such as DNSplatter, Gaussian Opacity Fields, and 2DGS, are missing. A table presenting geometric accuracy metrics (mesh extraction quality) across standard datasets like Tanks and Temples, DTU, or ETH3D would significantly strengthen the paper.
- Incomplete Mesh Extraction Evaluation: The current mesh extraction evaluation is limited to planar surfaces. To provide a comprehensive assessment of geometric accuracy for the entire scene, the authors should apply and evaluate a robust mesh extraction method, potentially leveraging the approach demonstrated in 2DGS, which appears straightforward to implement.

---

> ### Author Rebuttal · Authors · 2025-07-31
>
> We thank you for acknowledging the novelty of our hybrid 2D+3D Gaussian representation. Below, we clarify the motivation behind our design and provide additional evidence on how our method achieves improved geometric accuracy.
>
> **Limited justification for hybrid approach.**
> Our primary motivation is to achieve both photometric and geometric accuracy in indoor scenes, which often contain large, weakly textured planar surfaces. Optimizing purely for photometric loss and without geometric priors, as done in 3DGS and 2DGS, often fails to capture correct geometry in these areas, particularly with specular or uniform textures like walls (e.g. see DN-Splatter [1] within which the caption of Figure 3 clearly states that “The Gaussian based methods Splatfacto, SuGaR, and 2DGS are trained on only photometric losses and thus severely struggle to capture the scene geometry in low texture environments").
>
> **Q: Why not use depth (e.g. DN-Splatter) or normal supervision?**
> Note that DN-Splatter assumes depth input from a sensor, not only rendering the problem of 3D reconstruction significantly easier, but limiting applicability with data acquired by a limited class of devices. More specifically, only Apple has continued shipping LiDAR sensors in consumer smartphones, with their top-of-the-line “Pro Max” trim carrying them. Samsung, back in 2019–2020 experimented with rear-facing time‑of‑flight (ToF) sensors on some Galaxy S10 5G, S20, and S20+ models, but these were not Apple‑style LiDAR and Samsung later discontinued them. We argue that investigating techniques that do not rely/require depth input is critical, as it allows a broader consumer base to reconstruct 3D structures with photometric supervision alone.
>
> Monocular depth predictors are a possible alternative, but they can only produce view-inconsistent results, limiting their effectiveness for supervision. For example, the recent Video Depth Anything [2] states how depth consistency in video still presents a challenge, and non-sequential multi-view data (a less structured generalization of video input) is not even considered.
>
>
> In contrast, our method adds a simple but strong planar prior, requiring minimal user input and leveraging RANSAC to robustly detect and propagate 3D plane assignments. For planar regions, where the geometry is already well-defined, this offers a lightweight alternative tailored to the dominant structures in the indoor scenes.
>
> **Q: Why not use only 2DGS?**
> While 2DGS offers improved geometry over 3DGS, it does so at a significant photometric cost. For example, as shown in Figure 4, 2DGS drops PSNR by ~1.7dB, while our hybrid method limits the drop to ~0.2dB, while simultaneously achieving superior depth quality. 3D Gaussians better capture photometric detail and complex geometry, while 2D Gaussians are ideal for modeling planar regions. Our hybrid design combines these strengths, yielding better geometry on flat surfaces without significantly compromising overall image quality. The remaining slight drop in performance wrt. 3DGS (0.2dB) is caused by the limited expressive power of spherical harmonics, as once one imposes a “surface” (vs. volume) reconstruction, complex view-dependent effects cannot any longer be represented by volumetric effects.
>
> **Insufficient experimental validation (i.e. our new experiments).**
> Our experimental evaluation focuses on both photometric accuracy (rendering color error) and geometric accuracy (rendered depth error), which are direct outputs of general-purpose 3D reconstruction pipelines. These metrics are reported on ScanNet++, a dataset well aligned with our goal of accurate indoor scene reconstruction.
> Tangentially, we include mesh accuracy comparisons to prior plane-based 3D reconstruction methods, which generate meshes for the planar parts of a scene. However, we acknowledge the value of extracted mesh accuracy comparisons with non-planar baselines as additional evidence, and thank the reviewer for the suggestion.
>
>
> Among the suggested datasets, only ETH3D features structured planar geometry, while Tanks and Temples and DTU are primarily “object-centric” and less relevant to our use case. DTU consists of small objects that do not contain a sufficient number of planes to make our method beneficial. Tanks and Temples scenes contain not enough scenes of interest as it is mostly focused on complex non-planar geometry.
>
>
> Due to the limited time and compute budget, we provide additional mesh accuracy results on a selection of scenes from ETH3D (using the same mesh extraction strategy as 2DGS). The outdoor scenes were selected to contain parts of the roads and buildings to show the generalization of the method; they are also sparse (<40 train images) and unbounded. The selected indoor scene is similar to the most challenging scene from ScanNet++ we evaluated on in the paper to show generalization on indoor scenes. Note that we cannot evaluate DN-Splatter on ETH3D, as in this dataset the provided depth maps are used to create the ground truth mesh.
>
> Our hybrid method yields competitive geometry results in these scenes, with higher photometric accuracy, further validating the benefit of incorporating planar priors. We leave the fusing of the detected planar geometry into the mesh for future work, but this would simply further improve the mesh quality results.
>
> `Electro` (outdoor)
> |       | PSNR↑ | LPIPS↓ | SSIM↑ | Chamfer↓ | F1↑    |
> |-------|-------|--------|-------|----------|--------|
> | 3DGS  | 16.45 | 0.38   | 0.72  | 0.7282   | 0.0923 |
> | 2DGS  | 16.40 | 0.41   | 0.72  | 0.6156   | 0.2251 |
> | GOF   | 17.34 | 0.36   | 0.71  | 1.4481   | 0.2305 |
> | Ours  | 18.72 | 0.31   | 0.75  | 0.4922   | 0.3170 |
>
> `Terrace` (outdoor)
> |       | PSNR↑ | LPIPS↓ | SSIM↑ | Chamfer↓ | F1↑    |
> |-------|-------|--------|-------|----------|--------|
> | 3DGS  | 20.77 | 0.27   | 0.78  | 0.5778   | 0.0385 |
> | 2DGS  | 20.82 | 0.29   | 0.79  | 0.3411   | 0.3230 |
> | GOF   | 20.80  | 0.27   | 0.75  | 0.2118   | 0.3901 |
> | Ours  | 22.57 | 0.22   | 0.81  | 0.2795   | 0.4862 |
>
> `Delivery area` (indoor)
> |       | PSNR↑ | LPIPS↓ | SSIM↑ | Chamfer↓ | F1↑    |
> |-------|-------|--------|-------|----------|--------|
> | 3DGS  | 19.48 | 0.29   | 0.83  | 0.5945   | 0.0727 |
> | 2DGS  | 19.26 | 0.35   | 0.81  | 0.3200   | 0.1783 |
> | GOF   | 19.40 | 0.33   | 0.79  | 0.3044   | 0.2740 |
> | Ours  | 22.56 | 0.21   | 0.87  | 0.2305   | 0.3281 |
>
> - `Electro` the most challenging as it contains sparse unbounded views of the cloudy sky, where Gaussian Opacity Fields struggles to extract a correct mesh, while 2DGS mesh extraction uses depth truncation, so it filters out the sky.
> - `Terrace`  the geometry is dominated by a long building in which the default depth truncation parameters for 2DGS and our method cut off the geometry, while Gaussian Opacity Fields mesh extraction extracts that, yet not precisely, while the planar geometry in our method is more precise as evidenced by F1-score.
> - `Delivery area` indoor scene where our method outperforms the benchmarks that use photometric supervision.
>
>
> **Incomplete mesh extraction evaluation.**
> Below, we follow reviewer’s suggestion and further provide quantitative results on the extracted meshes of ScanNet++ compared to 2DGS, 3DGS, Gaussian Opacity Fields and DN-Splatter.
> We provide the results for 11 scenes for the methods from the paper and additional results on the one scene for Gaussian Opacity Fields and DN-Splatter due to the limited time and available compute budget.
> We achieve competitive performance over all baselines, trading off geometric accuracy and rendering quality. Note that DN-Splatter uses sensor depth as input, which drastically improves the quality of mesh, however, at the cost of render quality.
>
>
> `ScanNet++ (11 scenes)`
>
> |              | PSNR↑ | LPIPS↓ | SSIM↑ | Chamfer↓ | F1↑    |
> |--------------|-------|--------|-------|----------|--------|
> | 3DGS         | 27.09 | 0.20   | 0.89  | 0.1972   | 0.3440 |
> | 2DGS         | 25.56 | 0.24   | 0.88  | 0.1725   | 0.4943 |
> | Ours         | 27.01 | 0.21   | 0.89  | 0.1467   | 0.5666 |
>
>
> `ScanNet++ (0a7cc12c0e)`
>
> |                 | Input  | PSNR↑ | LPIPS↓ | SSIM↑ | Chamfer↓ | F1↑    |
> |-----------------|--------|-------|--------|-------|----------|--------|
> | 3DGS            | RGB    | 30.19 | 0.16   | 0.94  | 0.1570   | 0.2810 |
> | 2DGS            | RGB    | 24.19 | 0.27   | 0.89  | 0.1512   | 0.4117 |
> | GOF             | RGB    | 26.98 | 0.20   | 0.92  | 0.1173   | 0.4861 |
> | DN-Splatter     | RGB+D  | 26.73 | 0.10   | 0.92  | 0.0801   | 0.8777 |
> | Ours            | RGB    | 28.16 | 0.20   | 0.93  | 0.1178   | 0.4975 |
>
>
> **Q: What RANSAC did the authors use to get the plane parameters?**
> We used a custom pytorch implementation of the RANSAC algorithm as described in classical computer vision educational material.
>
> **Bibliography.**
> - [1] Turkulainen, Matias, et al. "Dn-splatter: Depth and normal priors for gaussian splatting and meshing." 2025 IEEE/CVF Winter Conference on Applications of Computer Vision (WACV). IEEE, 2025.
> - [2] Chen, Sili, et al. "Video depth anything: Consistent depth estimation for super-long videos." Proceedings of the Computer Vision and Pattern Recognition Conference. 2025.

---

> > ### Comment · Reviewer_bJrA · 2025-08-04
> >
> > - "Note that we cannot evaluate DN-Splatter on ETH3D, as in this dataset the provided depth maps are used to create the ground truth mesh." -> DN-Splatter works well with predicted depth. It provides code for that. The author could simply compare it to that. Same on the other datasets.
> > - These accuracies are hard to judge without knowing the F1 score's threshold, but they seem incorrect. What meshing did the authors use for 3DGS? I actually have results for both 3DGS and GOF on ETH3D and this is how they look with 1 cm, 3 cm, and 5 cm F1 thresholds:
> >
> > Electro:
> > 3DGS: 14,5 | 27,2 | 49,8
> > GOF: 13,5 | 20,6 | 34,9
> >
> > Terrace
> > 3DGS: 10,5 | 25,1 | 60,7
> > GOF: 35,2 | 50,7 | 73,4
> >
> > Delivery area
> > 3DGS: 10,8 | 23,0 | 51,5
> > GOF: 19,0 | 30,7 | 52,1
> >
> > 3DGS in the authors' experiments is much worse than it is supposed to be. I was using TSDF fusion from 2DGS for the meshing. Thus, something seems incorrect in these experiments. I suggest the authors to double-check.
> >
> > Due to these seemingly incorrect results (or at least not satisfactorily explained) and that DN-Splatter is missing with predicted depth, I will keep my original Reject rating.

---

> > > ### Author Response · Authors · 2025-08-05
> > > **Response for the Reviewer bJrA**
> > >
> > > Thanks for acknowledging the rebuttal.
> > >
> > > **3DGS meshing**
> > > - The F1 metric in the results is reported at the 5cm threshold, using the metric computation script similar to DNSplatter.
> > > - We used the most recent version of the official 3DGS GitHub repository to run 3DGS experiments.
> > > - Then we applied TSDF fusion to do the meshing, taking the code from 2DGS, with default voxelization and truncation parameters, that adjust to the scene extent.
> > > - For depth computation of the 3DGS, we used the official rasterizer, which computes expected ray termination as depth. This might be the source of difference in metrics that Reviewer **bJrA** provides, please let us know if you are using a different method for computing depth. We used a similar rasterizer for the meshing results in our method, with expected ray termination as depth.
> > > - For 2DGS we similarly used mean depth as is done by default in the official codebase. An alternative method provided in the paper is using median depth. We will provide the updated results with **median** depth for 3DGS/2DGS and ours.
> > > - For GOF we used the lambda_distortion parameter set for indoor/outdoor scenes as in the paper and default meshing.
> > >
> > > **DNSplatter mono-depth**
> > > - Thanks for pointing out that DNSplatter works with monocular depth estimation models, the mono-depth evaluation pipeline was not clear from the provided README, we will work on providing a comparison with predicted depth.
> > > - While we agree that such a comparison would strengthen the discussion, we would like to reiterate our motivation. While depth-based methods like DNSplatter aim for dense reconstruction, our method targets a specific failure mode of 3DGS: distorted planar surfaces. We introduce simple planar priors to improve geometry in these regions without relying on full-scene depth prediction. This offers a complementary, geometry-driven alternative that improves accuracy on challenging surfaces while retaining the rendering quality of 3DGS.

---

> > > > ### Author Response · Authors · 2025-08-06
> > > >
> > > > The code available online does not reproduce the F1 scores reported by the reviewer. In the following evaluation, we computed the F1 metric at a 5 cm threshold on full meshes using the median depth estimate, rather than the expected ray termination as in default settings (i.e., average depth). For the ETH3D dataset, we use every 8th image as a test image. We found that the `Electro’ scene is unbounded and setting the far plane to 10 performs better for GOF than the default setting.
> > > >
> > > > For metrics computation we use the scripts from DNSplatter, for predicted ETH3D meshes we sample uniformly by mesh area as in DNSplatter code for ScanNet++ evaluation, then from predicted and from GT pointclouds of ETH3D we sample 2M points randomly and compute the metrics. We use the routine from DNSplatter to compute Chamfer and F1 score.
> > > >
> > > > `Electro` (outdoor)
> > > > |            | PSNR↑ | LPIPS↓ | SSIM↑ | Chamfer↓ | F1↑    |
> > > > |------------|-------|--------|-------|----------|--------|
> > > > | 3DGS       | 16.45 | 0.38   | 0.72  | 0.6524   | 0.2511 |
> > > > | 2DGS       | 16.40 | 0.41   | 0.72  | 0.5873   | 0.2570 |
> > > > | GOF        | 17.34 | 0.36   | 0.71  | 0.5371   | 0.2991 |
> > > > | Ours       | 18.72 | 0.31   | 0.75  | 0.4062   | 0.3009 |
> > > >
> > > > `Terrace` (outdoor)
> > > > |            | PSNR↑ | LPIPS↓ | SSIM↑ | Chamfer↓ | F1↑    |
> > > > |------------|-------|--------|-------|----------|--------|
> > > > | 3DGS       | 20.77 | 0.27   | 0.78  | 0.3258   | 0.4517 |
> > > > | 2DGS       | 20.82 | 0.29   | 0.79  | 0.3312   | 0.4036 |
> > > > | GOF        | 20.80 | 0.27   | 0.75  | 0.2107   | 0.4045 |
> > > > | Ours       | 22.57 | 0.22   | 0.81  | 0.1480   | 0.5033 |
> > > >
> > > > `Delivery area` (indoor)
> > > > |            | PSNR↑ | LPIPS↓ | SSIM↑ | Chamfer↓ | F1↑    |
> > > > |------------|-------|--------|-------|----------|--------|
> > > > | 3DGS       | 19.48 | 0.29   | 0.83  | 0.3064   | 0.2335 |
> > > > | 2DGS       | 19.26 | 0.35   | 0.81  | 0.3265   | 0.2366 |
> > > > | GOF        | 19.40 | 0.33   | 0.79  | 0.2939   | 0.3131 |
> > > > | Ours       | 22.56 | 0.21   | 0.87  | 0.1825   | 0.3313 |
> > > >
> > > >
> > > > **Given these efforts, we would kindly ask the reviewer to point us to a peer reviewed source and published code that contains instructions on how to reproduce their reported numbers.**
> > > > We would also like to emphasize that our code, along with detailed evaluation scripts, will be made publicly available to ensure full transparency and reproducibility.
> > > >
> > > > For DNSplatter using mono-depth, we provide results on the 'delivery area’ scene with the proposed meshing method with the same parameters as for others (e.g. depth truncation and voxel size), and their default depth estimate (mean depth). The released codebase does not support multiple camera models (different camera intrinsics) for aligning mono-depth to SfM points, therefore we cannot easily report the metrics for `Electro’ and `Terrace’ scenes.
> > > >
> > > > `Delivery area` (indoor)
> > > > |                         | PSNR↑ | LPIPS↓ | SSIM↑ | Chamfer↓ | F1↑    |
> > > > |-------------------------|-------|--------|-------|----------|--------|
> > > > | DNSplatter (mean depth) | 19.56 | 0.24   | 0.77  | 0.2488   | 0.2516 |
> > > > | Ours (mean depth)       | 22.56 | 0.21   | 0.87  | 0.2305   | 0.3281 |

---

> ### Comment · Reviewer_bJrA · 2025-08-07
>
> I thank the authors for the new experiments. These are the trends that I have seen as well, with 3DGS often outperforming or being on par with methods designed for improving geometry. I am happy with the results and willing to improve my rating.

---

> > ### Author Response · Authors · 2025-08-07
> > **Answer to bJrA**
> >
> > Thank you for the thoughtful suggestions and follow-up discussion. We are glad to hear that the additional evaluations helped contextualize the results, and we plan to incorporate the key findings from this exchange into the final version of our paper.
> >
> > As an additional verification step, we provide below an independent third-party evaluation of GOF on the `Delivery area' scene. These results were obtained using the parameters suggested by the paper, and were reproduced by an external expert unaffiliated with our author team. The reported F1 metric at the 5 cm threshold (0.05 in the table) is 0.3209, which closely aligns with our previously reported value of 0.3131.
> > We sincerely thank the reviewer again for the constructive dialogue, which we believe has contributed meaningfully to strengthening the paper.
> >
> >
> > F1 scores
> > | Tolerance      | 0.01     | 0.02     | 0.05     | 0.1      | 0.2      | 0.5      |
> > |----------------|----------|----------|----------|----------|----------|----------|
> > | Delivery area  | 0.069137 | 0.149012 | 0.320927 | 0.503294 | 0.699391 | 0.894986 |

---

### Note · Authors · 2025-08-13

We thank the reviewers for the constructive dialogue and the opportunity to present the additional evaluations.


Reconstructing indoor scenes with many planar, low-texture surfaces is challenging. Most 3DGS- or 2DGS-based methods trade appearance for geometry in these regions, producing either photorealistic renders with noisy geometry or accurate shapes lacking view-dependent effects and fidelity. Our work targets **both** geometric accuracy and photorealism.


Our main contribution is incorporating planar priors into 3DGS, enabling accurate reconstruction of flat scene regions. We detect 3D planes and model them with 2D Gaussians, and represent the rest of the scene with 3D Gaussians, leveraging their strength for photorealistic rendering. We achieve SoTA depth accuracy and photometric error on common indoor scenes from SceneNet++ and ScanNet, and show generalizability to additional indoor **and** outdoor scenes on ETH3D. Our method also delivers competitive full mesh recovery compared to surface reconstruction methods, confirming our ability to recover accurate geometry without compromising appearance.


We appreciate reviewer **bJrA** ’s detailed suggestion to evaluate full mesh recovery and metrics. We’re glad to have reached consensus on the reported metrics and to demonstrate competitive results on this application compared to the SoTA surface reconstruction methods. We also hope this addresses concerns raised by reviewer **wJbn**, who recommended comparisons to PGSR on the same application, which we have included.


We are committed to incorporating the insights and additional experiments from the rebuttal phase into the final paper. Additionally, we will make sure to improve the figures’ image quality, which was previously reduced due to the submission file size limit.


We once again thank the reviewers and the AC for their valuable time and their insightful comments.

---

### Decision · Program_Chairs · 2025-09-17

**Decision:**

Accept (poster)

**Comment:**

After a strong rebuttal, all reviewers provided positive final recommendations (with three borderline accepts and one accept). The AC agrees with the reviewers that the combination of 2D and 3D Gaussian representations is interesting, and that the evaluations sufficiently demonstrate its effectiveness. The AC recommends acceptance.

Please include the additional evaluations in the revision and address the misleading points noted by the reviewers.